# The purine metabolite inosine monophosphate accelerates myelopoiesis and acute pancreatitis progression

Xiao-Min Luo[1,7], Sin Man Lam [2,7], Yuan Dong[1,7], Xiao-Juan Ma[3], Cen Yan[1], Yue-Jie Zhang[1], Yu Cao[1], Li Su[4], Guotao Lu[5], Jin-Kui Yang[6], Guanghou Shui [2✉] & Ying-Mei Feng [1✉]

Hyperglycemia-induced myelopoiesis and atherosclerotic progression occur in mice with type I diabetes. However, less is known about the effects of metabolites on myelopoesis in type 2 diabetes. Here, we use fluorescence-activated cell sorting to analyze the proliferation of granulocyte/monocyte progenitors (GMP) in db/db mice. Using targeted metabolomics, we identify an increase in inosine monophosphate (IMP) in GMP cells of 24-week-old mice. We show that IMP treatment stimulates cKit expression, ribosomal S6 activation, GMP proliferation, and Gr-1$^+$ granulocyte production in vitro. IMP activates pAkt in non-GMP cells. In vivo, using an established murine acute pancreatitis (AP) model, administration of IMP-treated bone marrow cells enhances the severity of AP. This effect is abolished in the presence of a pAkt inhibitor. Targeted metabolomics show that plasma levels of guanosine monophosphate are significantly higher in diabetic patients with AP. These findings provid a potential therapeutic target for the control of vascular complications in diabetes.

[1] Department of Science and Development, Beijing Youan hospital, Capital Medical University, 100069 Beijing, China. [2] State Key Laboratory of Molecular Developmental Biology, Institute of Genetics and Developmental Biology, Chinese Academy of Sciences, 100101 Beijing, China. [3] Center of Basic Medical Research, Institute of Medical Innovation and Research, Peking University Third Hospital, 49 North Garden Road, Haidian District, 100191 Beijing, China. [4] Neuroscience Research Institute, Peking University Center of Medical and Health Analysis, Peking University, 100191 Beijing, China. [5] Pancreatic Center, Department of Gastroenterology, Affiliated Hospital of Yangzhou University, Yangzhou University, 225099 Yangzhou, China. [6] Department of Endocrinology, Beijing Tongren Hospital, Capital Medical University, Beijing, China. [7] These authors contributed equally: Xiao-Min Luo, Sin Man Lam, Yuan Dong. ✉email: ghshui@genetics.ac.cn; yingmeif13@sina.com

In addition to hyperglycemia and insulin insufficiency, patients with type 2 diabetes mellitus (T2DM) often present with metabolic disorders, chronic inflammation and vascular pathogenesis. The inflammatory response is often activated in the adipose tissues of obese subjects, leading to the proliferation of proinflammatory Th1 and Th17 CD4[+] T cells and M2 macrophage polarization[1]. As the disease progresses, chronic inflammation is maintained and manifests as an increased myeloid cell number and a decreased lymphoid cell number, a process called skewed myelopoiesis. Skewed myelopoiesis has been demonstrated in streptozotocin-induced type 1 diabetic mice[2,3], obese db/db mice[4], and T2DM patients[5]. The inflammatory cells, together with proinflammatory cytokines, chemokines, and enzymes, reinforce the progression of diabetes and the resulting vascular complications.

The underlying mechanism of hyperglycemia- and hypercholesterolemia-induced myelopoiesis is well established. Studies using scavenger receptor class B type I knockout and apolipoprotein E (apoE) knockout mice models have demonstrated that hypercholesterolemia impairs cholesterol homeostasis and stimulates the proliferation of hematopoietic stem and progenitor cells (HSPCs), resulting in myeloid expansion and the progression of atherosclerosis[6,7]. These studies further showed that, when wild type (WT) mice were injected with streptozotocin to induce type 1 diabetes, hyperglycemia did not alter HSPC frequency; instead, it elevated the expression of the receptor for advanced glycation end production (RAGE) in common myeloid progenitor cells (CMP). In addition, hyperglycemia triggered the expression of GLUT1 in neutrophils, resulting in increased glucose uptake and S100A8/A9 production[8]. The binding of S100A8/A9 to RAGE caused increased production of colony-stimulating factors in CMP, leading to granulocyte-monocyte progenitor (GMP) proliferation, myelopoiesis and atherosclerotic plaque progression[9]. In this study, we interrogated the role of other metabolites that result from the altered metabolic state in T2DM diabetes, and investigated whether other metabolites could also modulate the proliferation of myeloid progenitor cells, resulting in myelopoiesis and strengthened inflammation in diabetic vascular complication.

To investigate this hypothesis, we studied db/db mice at 8 and 24 weeks old. We used fluorescence-activated cell sorting (FACS) to analyze the frequency and cell cycle of GMPs. Furthermore, we used metabolomics based on liquid-chromatography mass spectrometry (LC/MS) to comprehensively explore the metabolome of the isolated GMPs. The components of the bone marrow cell (BMC) microenvironment were also analyzed by LC/MS.

To dissect the vascular and pathology of skewed myelopoiesis, we applied a diabetes-associated acute pancreatitis (AP) model. Previous studies have produced compelling evidence to show that diabetes is not only an independent risk factor but can correlate with the severity of AP[10,11]. We therefore injected mice with caerulein-induced AP with BMCs from db/db mice at 8 and 24 weeks old in order to evaluate the effects of metabolites on GMP proliferation and AP progression.

## Results

### Diabetic db/db mice develop more severe acute pancreatitis.
24-week-old db/db mice showed impaired glucose tolerance and an increased white blood cell count compared with mice at 8 weeks (Fig. 1a, b). The percentages of granulocytes and monocytes among white blood cells were 29.4% and 7.4%, respectively, in mice at 8 weeks old, but increased to 38.8% and 9.0% in 24-week-old mice (granulocytes: $p = 0.033$; monocytes: $p = 0.063$) (Fig. 1c). In contrast, the percentage of lymphocytes was reduced by 11.0% in db/db mice at 24 weeks old compared

with 8-week-old mice (24 weeks: $52.2 \pm 5.0\%$; 8 weeks: $63.2 \pm 2.2\%$, $p = 0.024$, $n = 11–28$).

Previously, we showed that injection of BMCs from db/db mice into recipients with caerulein-induced AP resulted in more severe damage than injection of WT BMCs[12]. In this study, we used the same AP model to further compare the disease severity after injection of BMCs isolated from db/db mice at different ages. When BMCs were introduced to WT mice with caerulein-induced AP, we observed increased plasma amylase and lipase activity in the recipients that received BMCs isolated from 24-week-old db/db mice, compared with those that were injected with BMCs from younger mice (Fig. 1d, e). Similarly, the total and individual severity scores were higher in the recipients that received BMCs isolated from older db/db mice than BMCs from younger mice (Fig. 1f–h). We then stained pancreatic sections with an anti-myeloperoxidase (MPO) antibody and observed that the number of infiltrated MPO[+] cells was increased by 2.2-fold in recipients injected with BMCs from 24-week-old mice compared with BMCs from 8-week-old mice (321.6 ± 37.8 cells per microscopy field vs. 701.8 ± 79.9 cells per microscopy field, $n = 9–10$, $p = 0.0007$) (Fig. 1i, j).

### Granulocyte/monocyte progenitor proliferation in BM of diabetic db/db mice.
In order to determine whether the proliferation of HSPCs was caused by skewed myelopoiesis in diabetic db/db mice, we stained BMCs with a panel of markers for analysis by FACS. The frequency of lineage[−]Sca-1[+]cKit[+] (LSK) HSPCs in peripheral blood and bone marrow (BM) was similar in db/db mice at both 8 and 24 weeks (blood: $0.071 \pm 0.013\%$ vs. $0.077 \pm 0.013\%$, $p = 0.76$, $n = 6–9$; BM: $0.095 \pm 0.066\%$ vs. $0.098 \pm 0.006\%$, $p = 0.71$, $n = 17–20$) (Fig. 2a). However, the percentage of CMP, GMP, and megakaryocyte-erythroid progenitors (MEP) were all increased in BMCs from db/db mice at 24 weeks old compared with BMCs from mice at 8 weeks old (% CMP: $0.070 \pm 0.004\%$ vs. $0.088 \pm 0.006\%$, $p = 0.013$, $n = 12–22$; % GMP: $0.653 \pm 0.023\%$ vs. $0.831 \pm 0.066\%$, $p = 0.003$, $n = 12–24$; % MEP: $0.580 \pm 0.041\%$ vs. $0.775 \pm 0.052\%$, $p = 0.007$, $n = 12–24$) (Fig. 2b, c). The frequency of CMP, GMP, and MEP in splenocytes from the mice was similar between the two age groups ($n = 5–6$, $p \geq 0.17$) (Fig. 2d–f). The FACS analysis of HSPCs, CMP, GMP, and MEP in BMCs is shown in Fig. 2g.

To dissect the proliferative state of GMP, we injected bromodeoxyuridine (BrdU) peritoneally into db/db mice. We then used FACS to demonstrate that 30.5% of GMP in 8-week-old db/db mice incorporated BrdU; however, the incorporation increased significantly to 38.3% in db/db mice at 24 weeks old ($n = 9–12$, $p = 0.023$) (Fig. 2h). In contrast to GMP, the percentages of proliferative CMP and MEP were consistent between db/db mice of the two age groups (% BrdU-incorporating CMP in CMP population: $29.3 \pm 1.8\%$ vs. $28.0 \pm 2.7\%$, $n = 9–22$, $p = 0.71$; % BrdU-incorporating MEP in MEP population: $33.7 \pm 2.6\%$ vs. $33.1 \pm 3.4\%$, $n = 9–12$, $p = 0.90$) (Fig. 2h). Representative FACS analyses are shown in Fig. 2i.

### Enriched purine metabolites in BM microenvironment of db/db mice at 24 weeks.
Our next aim was to determine whether the BM microenvironment was altered in db/db mice following diabetic progression. We performed component analysis on equal amounts of BM fluid using LC-MS. Samples of the BM fluid of wt mice at 8 and 24 weeks old were included as controls. We generated a heat map to directly compare the four experimental groups (Supplementary Fig. 1a). In comparison to 8-week-old db/db mice, 44 proteins were upregulated and 17 proteins were downregulated in 24-week-old db/db mice. These data showed no

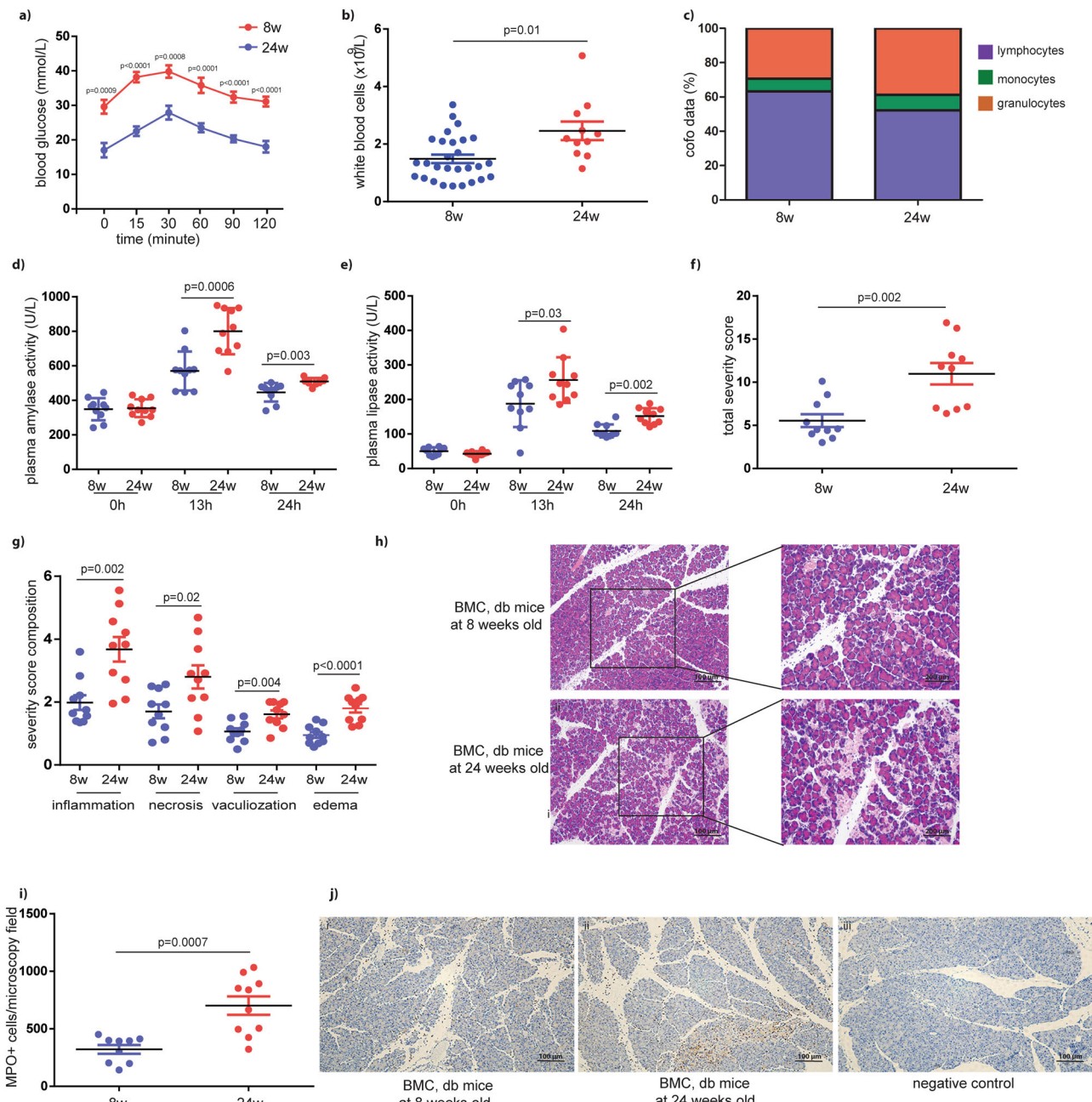

**Fig. 1 Diabetic db/db mice develop more severe acute pancreatitis. a** Glucose tolerance of db/db mice at 8 and 24 weeks old, $n = 7$–10. $p < 0.001$ for the entire time course. **b, c** White blood cell count and different white blood cell count, $n = 11$–28. **d, e** Serum amylase activity and serum lipase activity in mice with induced acute pancreatitis followed by injection of BMCs from db/db mice, $n = 10$. **f–h** The severity was assessed on H&E-stained sections by measuring inflammatory cell infiltration, acinar cell necrosis and vacuolization, and edema of the pancreas, scale bar = 100 μm or 200 μm, respectively. The total severity score is calculated as the sum of inflammation, necrosis, vacuolization, and edema, $n = 10$. **i, j** Paraffin sections were stained with an antibody against MPO. Representative images of MPO-stained sections in db/db mice are shown. Black arrows indicate MPO+ cells, scale bar = 100 μm. The number of MPO+ cells per microscopy field was estimated by ImageJ, $n = 10$. MPO, myeloperoxidase. Unpaired, 2-tailed Student's $t$ test for data with normal distribution. Non-parametric Mann–Whitney analysis was used for the data did not follow normal distribution.

significant difference in the abundance of glycolysis-related proteins between the two groups of mice ($p \geq 0.13$) (Supplementary Fig. 1b). In contrast, proteins that are involved in purine metabolism, including adenylosuccinate synthetase (ADSS) and IMP dehydrogenase 2 (IMPDH2), were present at higher levels in db/ db mice at 24 weeks old compared with mice at 8 weeks old. In WT mice, however, these proteins were present at comparable levels between mice at 8 and 24 weeks. Similar results were obtained using an ELISA assay (Supplementary Fig. 1c, d). We further observed that the concentration of guanosine monophosphate was significantly higher in the BM fluid and liver of db/db mice at 24 weeks old than db/db mice at 8 weeks old (BM fluid: 39.47. ± 4.75 pg/mL vs. 44.83 ± 3.14 pg/mL, $n = 9$–16, $p = 0.008$; liver: 46.58 ± 1.47 pg/mL vs. 51.05 ± 1.18 pg/mL, $n = 6$, $p = 0.04$) (Supplementary Fig. 1e, f). Together, these data suggest that purine metabolism may increase following diabetic progression.

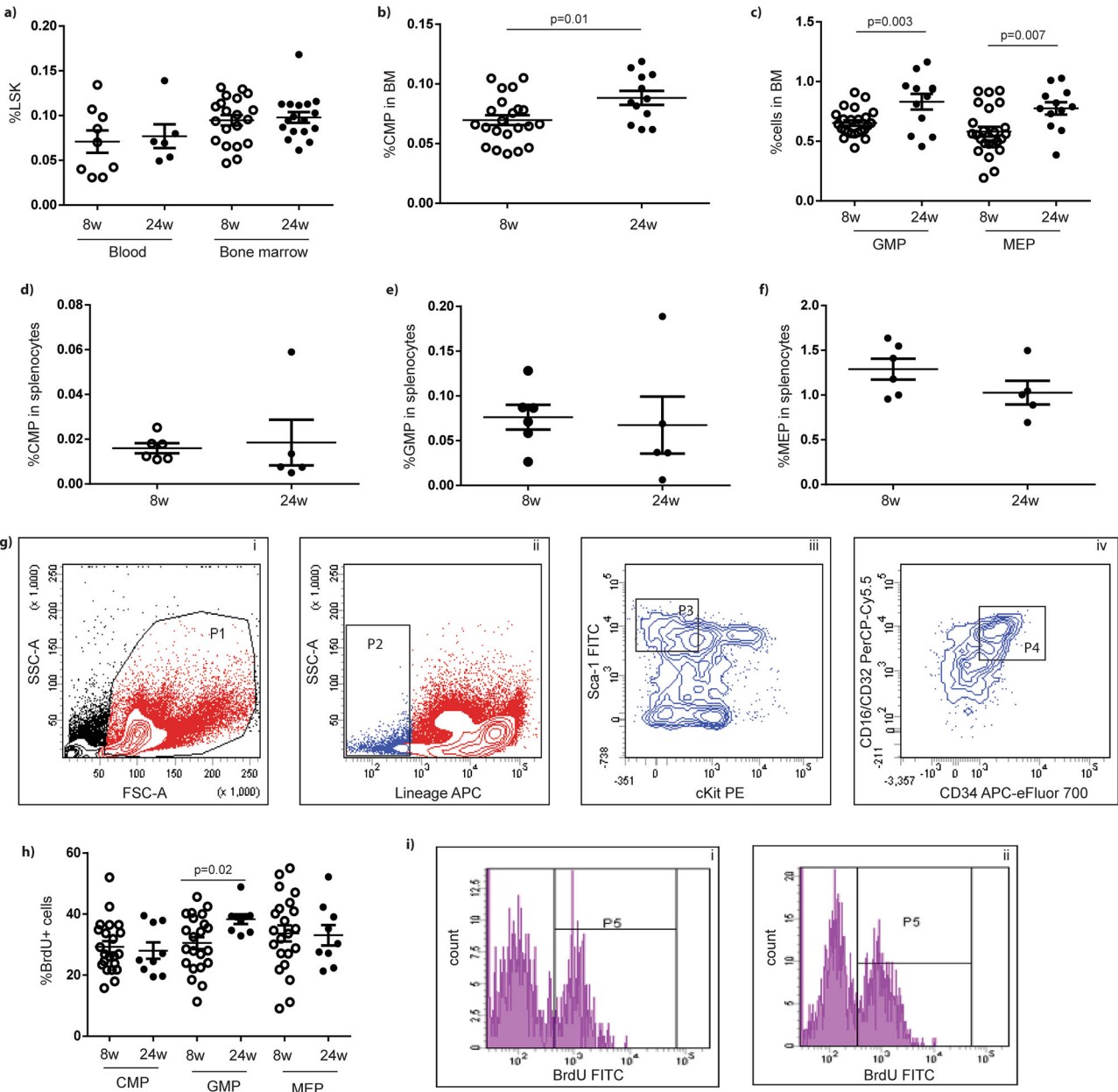

**Fig. 2 Proliferative status of granulocyte-monocyte progenitors in BM of diabetic db/db mice. a** Bone marrow LSK frequency of db/db mice at the age of 8 and 24 weeks old, $n = 14$–17. **b, c** Bone marrow CMP, GMP, and MEP cell frequency of db/db mice at the age of 24 weeks old, $n = 12$–24. **d**–**f** CMP, GMP and MEP cell frequency in splenocytes of db/db mice at both ages, $n = 5$–6. **g** FACS analysis of HSPCs, CMP, GMP, and MEP cells in BMC, $n = 12$–24. **h** Quantification of BrdU-incorporating CMP, GMP, and MEP in db/db mice at both ages, $n = 9$–12. **i** Representative FACS analysis of proliferative GMP population, $n = 9$–12. LSK, Lin- Sca-1+cKit+ cells. CMP, common myeloid progenitors. GMP, granulocyte-monocyte progenitors. MEP, meyakaryocyte-erythroid progenitors. BMC, bone marrow cells. Unpaired, 2-tailed Student's t test for data with normal distribution. Non-parametric Mann–Whitney analysis was used for the data did not follow normal distribution.

**Increased purine metabolites in GMP cells of db/db mice at 24 weeks old.** Next, we performed targeted metabonomics to interrogate the metabolic state of FACS-sorted GMP cells from db/db mice. A scheme of experimental design is shown in Fig. 3a. The intracellular levels of each metabolite in individual mice were quantified by metabonomics (Supplementary Table 1). The volcano plot illustrates the difference in concentration of each metabolite between the two groups of mice (Fig. 3b).

The presence of ribose-5-phosphate initiates purine metabolism, and guanosine triphosphate is one of the core metabolites downstream of the purine synthesis pathway[13]. The metabonomic data show that ribose-5-phosphate, guanosine triphosphate, and

inosine monophosphate were all present at significantly higher levels in GMP cells from 24-week-old mice than cells from 8-week-old mice, according to a Mann–Whitney U test (Fig. 3c–e). We observed no difference in the mRNA levels of ADSS, guanosine monophosphate reductase, or IMPDH2 in the FACS-sorted GMP cells between the two groups (Fig. 3f–i). Taken together, these results suggest that purine metabolism is elevated in GMP cells of db/db mice following the progression of diabetes.

**Inosine monophosphate treatment stimulates GMP cell proliferation and granulocyte production.** Within the purine metabolism pathway, the purine nucleotide inosine-5′-

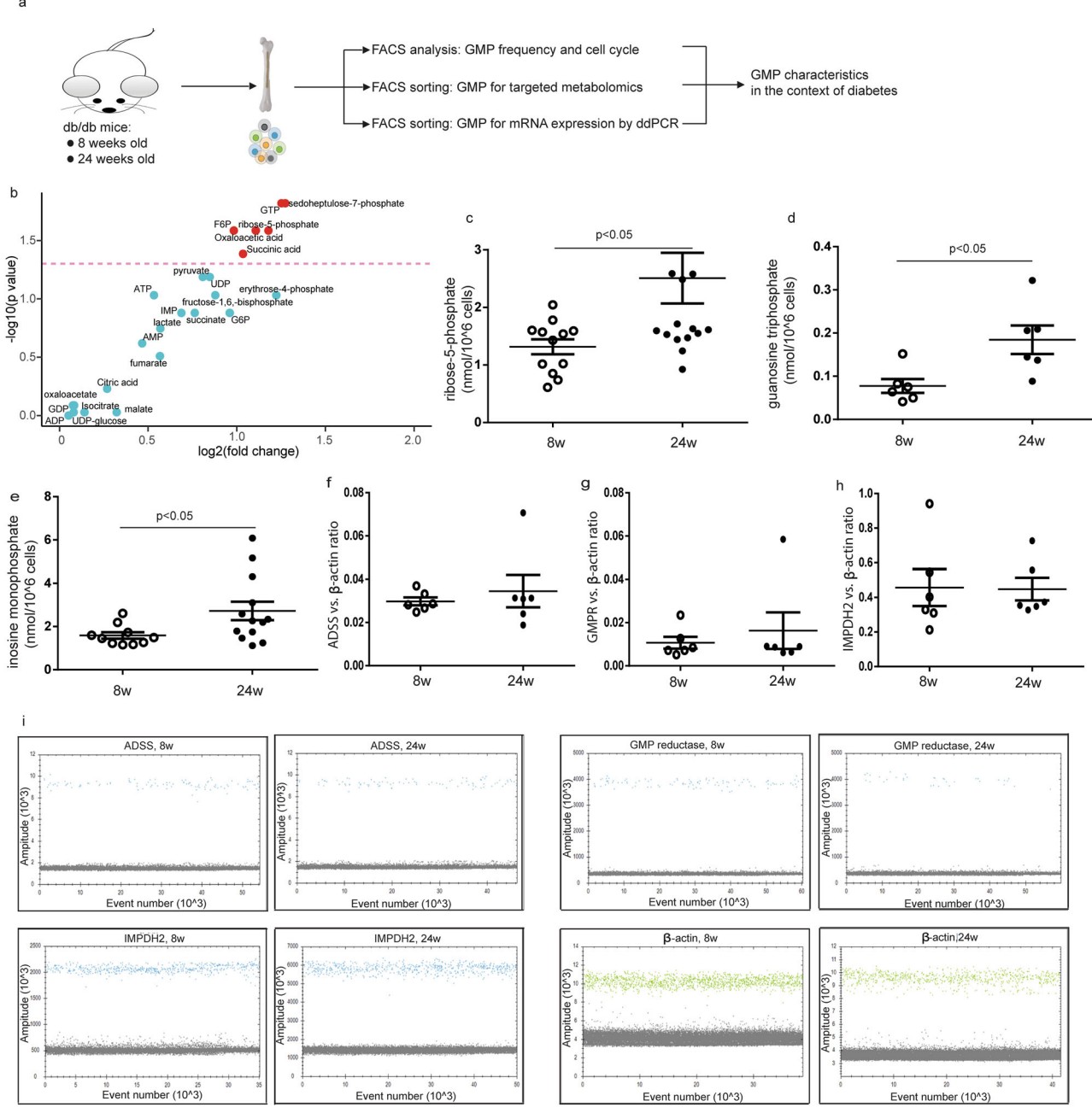

**Fig. 3 Increased purine metabolites in GMP cells of db/db mice aged at 24 weeks old.** The metabolic features of FACS-sorted GMP cells were detected by targeted metabonomics in db/db mice at the age of 8 and 24 weeks. **a** The scheme of the study design. **b** Volcano blot showing the difference of metabolite between two groups, $n = 6$–16. The pink line represented where $p$ value was 0.05. **c–e** Intracellular levels of ribose-5-phosphate, guanosine triphosphate, and inosine monophosphate in GMP cells of db/db mice. $n = 6$–14. **f–h** The mRNA expressions of ADSS, GMPR, and IMPDH2 were analyzed by ddPCR. $n = 6$. **i** Representative of ddPCR results using FACS-sorted GMP cells. ADSS, adenylosuccinate synthetase. Unpaired, 2-tailed Student's $t$ test for data with normal distribution. Non-parametric Mann–Whitney analysis was used for the data did not follow normal distribution. GMPR, guanosine monophosphate reductase. IMPDH2, IMP dehydrogenase 2.

monophosphate (IMP) plays a key role in the synthesis of guanosine monophosphate. To evaluate the impact of purine metabolites on GMP cells, lineage$^{-/low}$ cells were cultured in PBS, IMP, or guanosine triphosphate for 3 days. We observed that treatment with IMP enhanced cell proliferation in a dose-dependent manner; whereas, guanosine triphosphate treatment had no effect ($p < 0.0001$ for IMP; $p \geq 0.08$ for GTP) (Fig. 4a). After three days of growth, neither IMP nor guanosine triphosphate altered LSK cell production ($p \geq 0.24$ for all) (Fig. 4b). However, the frequency and absolute GMP cell number was

increased in IMP-treated cells compared with the control (GMP cell number at day 1: $1269.8 \pm 449.2$ vs. $1393.8 \pm 298.7$, $n = 6$, $p = 0.62$; GMP cell number at day 3: $1363.9 \pm 213.9$ vs. $2234.6 \pm 808.6$, $n = 7$, $p = 0.025$) (Fig. 4c, d). In contrast, guanosine triphosphate did not induce any change in GMP cell growth ($p \geq 0.47$) (Fig. 4c, d). To determine whether increased GMP production was due to enhanced proliferation, BrdU was added to the cells and incorporation of BrdU into GMP was measured by FACS. The addition of IMP resulted in a 1.3-fold increase in BrdU incorporation into GMP among the GMP

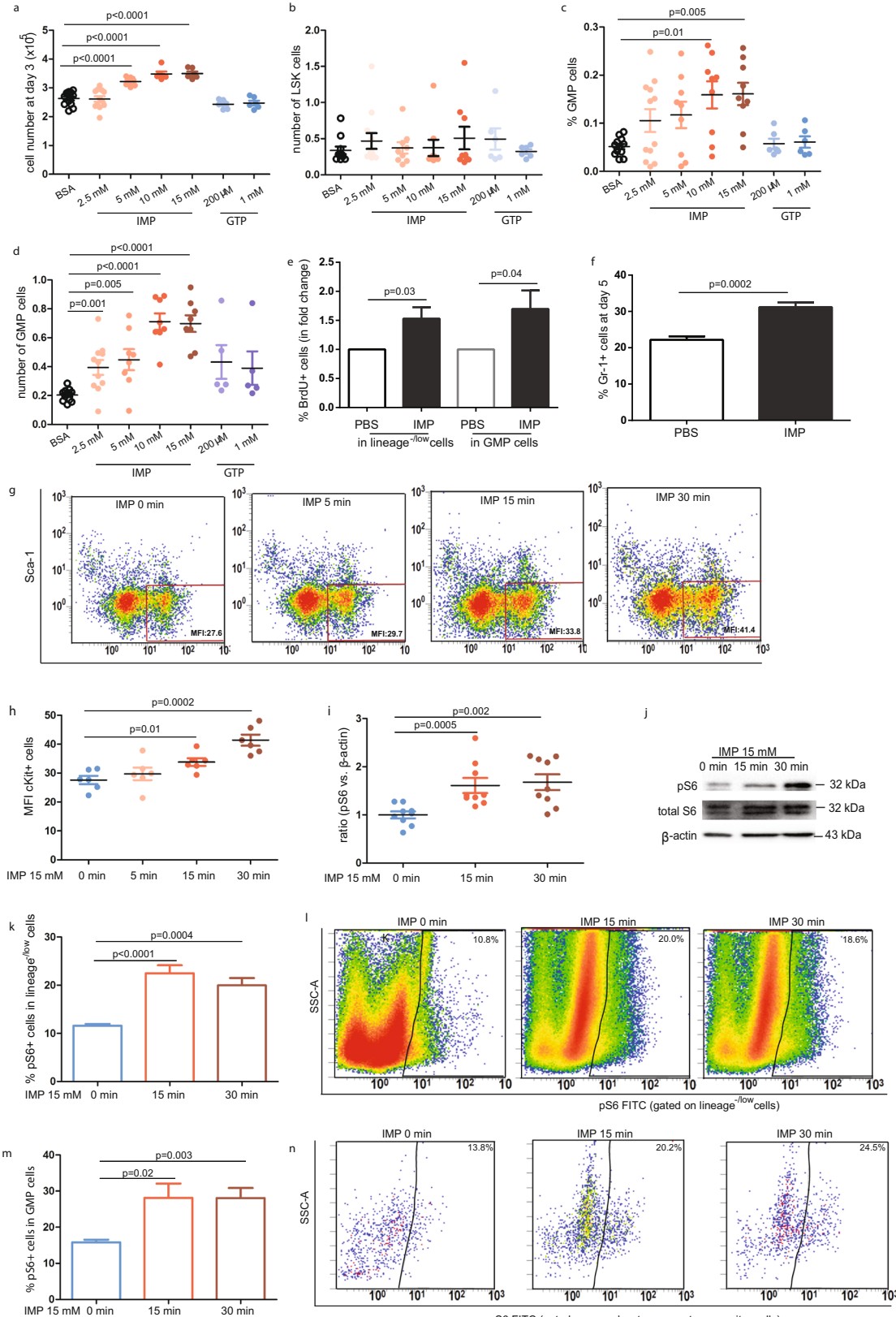

population (% BrdU+ GMP: $35.0 \pm 0.8\%$ vs. $44.8 \pm 1.1\%$, $n = 6$ for each group, $p < 0.0001$) (Fig. 4e). We also noted that the proportion of Gr-1+ cells was increased by 1.4-fold in IMP-treated cells compared with control cells (% Gr-1+ cells: $22.2 \pm 0.9\%$ vs. $31.2 \pm 1.3\%$, $n = 6$, $p < 0.001$) (Fig. 4f). We therefore concluded that IMP induces myeloid cell expansion.

We next used flow cytometry to show that the expression of cKit was increased 1.1-, 1.2-, and 1.5-fold in the membrane of lineage$^{-/low}$Sca-1$^{-}$ cells following transient exposure of the cells to IMP for 5, 15, and 30 min, respectively (mean fluorescence intensity (MFI): $27.6 \pm 1.4$ for 0 min, $29.7 \pm 2.2$ for 5 min, $33.8 \pm 1.3$ for 15 min, $41.4 \pm 1.9$ for 30 min; $n = 6$) (Fig. 4g, h).

**Fig. 4 Inosine monophosphate treatment stimulates GMP cell proliferation and Gr-1+ granulocyte production. a** Total cell number of after 3-day treatment of inosine monophosphate or guanosine triphosphate at different concentrations, $n = 6$–12. **b** Number of LSK cells after 3-day treatment of inosine monophosphate or guanosine triphosphate at different concentrations, $n = 6$–9. **c, d** The frequency and cell number of GMP after 3-day treatment of inosine monophosphate or guanosine triphosphate at different concentrations, $n = 5$–12. **e** Quantification of BrdU-incorporating lineage$^{-/low}$ or GMP cells after 3-day treatment of IMP analyzed by FACS, $n = 6$. **f** FACS analysis of Gr-1$^{+}$ granulocyte production after five days of IMP treatment. $n = 6$. **g** Quantification of cKit expression on lineage$^{-/low}$Sca-1$^{-}$ cells membrane after transient exposure of IMP for 5, 15 and 30 min, respectively, $n = 6$. **h** Representative analysis of cKit expression in lineage$^{-/low}$ cells. **i, j** Western blot of total S6 and phosphorylation of S6 in lineage$^{-/low}$ cells after exposure of IMP for 15 and 30 min, respectively. β-actin was used as loading control, $n = 9$. **k–n** Quantification of FACS data of pS6-positive cells in lineage$^{-/low}$ and GMP cells, respectively, $n = 5$–6. Representative FACS analysis of pS6 activation in lineage$^{-/low}$ and GMP cells (**k**) and (**m**). One-way ANOVA with Dunnett was used to compare groups against control.

Furthermore, we examined phosphorylation activation pathways using a Bio-Plex Pro™ Cell Signaling Assay. These data showed that the expression of phosphorylated ribosomal S6 protein was increased by 24.0-fold in lineage$^{-/low}$ cells after 15 min of exposure to IMP compared with control cells (MFI: $779.5 ± 98.5$ vs. $18798.0 ± 5984.8$, $p = 0.03$, $n = 6$). Western blots confirmed that ribosomal S6 protein phosphorylation was increased 1.6- and 1.7-fold after exposure of the lineage$^{-/low}$ cells to IMP for 15 and 30 min, respectively ($p ≤ 0.003$ for both, $n = 6$) (Fig. 4i, j). To further clarify the effect of IMP on the activation of ribosomal S6, freshly isolated BMCs were exposed to IMP and then stained with GMP cell markers and an antibody against phospho-ribosomal S6 (pS6) for FACS analysis. These data confirmed the activation of ribosomal S6 by IMP in both lineage$^{-/low}$ and GMP cells (% pS6+ cells in lineage$^{-/low}$ population: $11.6 ± 0.4\%$ for baseline, $22.5 ± 1.7\%$ for 15 min, $20.0 ± 1.5\%$ for 30 min; $p ≤ 0.0004$ for both; $n = 5$) (% pS6+ cells in GMP population: $15.8 ± 0.7\%$ for baseline, $28.1 ± 4.0\%$ for 15 min, $28.1 ± 2.8\%$ for 30 min; $p ≤ 0.02$; $n = 5$) (Fig. 4k–n). Representative FACS data showing the presence of pS6 in lineage$^{-/low}$ and GMP cells are shown in Fig. 4l, n.

Adenosine monophosphate (AMP) is an important regulator of the 5′ AMP-activated protein kinase pathway, which has been implicated in the pathogenesis of diabetes and diabetic complications. To determine whether AMP has any effect on GMP proliferation, lineage$^{-/low}$ cells were treated with different amounts of AMP for 3 days. Using FACS, we determined that neither GMP frequency nor myeloid cell production was altered by AMP treatment ($p ≥ 0.47$) (Supplementary Fig. 2a, b). Next, we treated lineage$^{-/low}$ cells with either PBS or 100 μM AMP for 7 days and then injected the cells intraperitoneally into wt recipients with established AP. AP mice that did not receive a cell injection were used as a control. Following cell injection, we observed that the activity levels of plasma amylase and lipase were similar between all groups ($p ≥ 0.10$ for all) (Supplementary Fig. 2c, d). Consistently, both the overall severity and each individual score were similar in AP mice that had or had not received a cell injection ($p = 0.42$ for total severity score; $p ≥ 0.25$ for individual severity scores) (Supplementary Fig. 2e–g).

**Inosine monophosphate treatment promotes mitochondrial transcription.** We next investigated whether IMP-induced GMP cell proliferation was mediated through effects on mitochondrial biogenesis. For this, we treated lineage$^{-/low}$ cells with IMP for 24 h and then looked at mRNA levels using qPCR. We observed that the levels of the mRNA transcripts encoding the nuclear respiratory factor 1 (NRF1) and the mitochondrial transcription factor A (mtTFA) were 3.2- and 2.5-fold higher compared with untreated cells (Fig. 5a). Moreover, western blot analysis showed a 1.7-fold increase in NRF1 protein following 24 h of IMP treatment (NRF1/β-actin ratio: $0.82 ± 0.07$ vs. $1.38 ± 0.15$, $p = 0.002$, $n = 15$). The protein expression of mtTFA was 2.5-fold higher in IMP-treated cells compared with untreated cells

(mtTFA/β-actin ratio: $0.36 ± 0.05$ vs. $0.91 ± 0.06$, $p < 0.0001$, $n = 27$). The expression of the purine synthesis enzyme phosphoribosylformylglycinamidine synthase (PFAS) was also observed to be increased by 1.7-fold in IMP-treated cells compared with control cells (PFAS/β-actin ratio: $0.56 ± 0.05$ vs. $0.98 ± 0.10$, $p = 0.0007$, $n = 18$) (Fig. 5b–e).

To examine whether IMP preferentially induced either ATP or GTP production, lineage$^{-/low}$ cells were exposed to IMP for 0, 4, or 8 h. Shortly after 4 h of incubation, we observed that IMP increased the activity of ADSS, ATPase and IMPDH2 in lineage$^{-/low}$ cells (ADSS activity: $p < 0.007$ for 4 and 8 h, $n = 6$; ATPase activity: $p = 0.017$ for 4 h and $p = 0.053$ for 8 h; IMPDH2 activity: $p < 0.05$ for 4 h, $n = 6$) (Fig. 5f–h). These enzymes are all critical for the production of both ATP and GTP. We then measured ATP production and observed that it was reduced by 20% after 24 h and by 45% after 72 h of IMP treatment ($p = 0.27$ for 24 h, $p = 0.006$ for 72 h, $n = 6$) (Fig. 5i). Conversely, GTP production was increased by 1.4-fold after 24 h and by 4.2-fold after 72 h following IMP treatment compared with a matched control ($p = 0.92$ for 24 h, $p = 0.003$ for 72 h, $n = 5$) (Fig. 5i). To further elucidate mitochondrial function, we then determined the oxygen consumption rate (OCR) in cultivated lineage$^{-/low}$ cells using a Seahorse bioscience XF96 extracellular flux analyzer. Consistent with the reduced ATPase activity and lowered ATP production, the OCR was significantly decreased in lineage$^{-/low}$ cells after 6 h of treatment with IMP (Fig. 5j, k).

**Inosine monophosphate treatment induces Akt activation in lineage$^{-/low}$ cells.** The phosphorylation of Akt (pAkt) is one of the main regulators of cellular proliferation. In this study, we interrogated whether the IMP-induced proliferation of GMP was mediated by the activation and phosphorylation of Akt. Western blots showed that IMP treatment indeed stimulated Akt activation in lineage$^{-/low}$ cells (Supplementary Fig. 3a, b). In contrast, when BMCs were treated with IMP for the same amount of time, Akt activation was detected in lineage$^{-/low}$ cells but not in GMP cells (Supplementary Fig. 3c). Lastly, we exposed lineage$^{-/low}$ cells to IMP in the presence or absence of the pAkt inhibitor MK2206 for 72 h. These data showed that the IMP-induced expression of mtTFA and NRF1 was abrogated by MK2206 (Supplementary Fig. 3d–f). Similarly, the increase in the frequency and number of GMP cells following treatment with IMP was abrogated by MK2206 (Supplementary Fig. 3g, h).

**Injection of inosine monophosphate-treated cells accelerates acute pancreatitis via Akt activation.** To clarify the effects of IMP and Akt activation on disease progression, we cultured lineage$^{-/low}$ cells with IMP in the presence or absence of MK2206 for 7 days, and then injected the cells peritoneally into wt mice with induced AP (Supplementary Fig. 4a). The results were similar to the injection of BMCs (Fig. 1): plasma levels of lipase and amylase activity were elevated in mice that were injected with IMP-treated cells compared with the control (plasma lipase

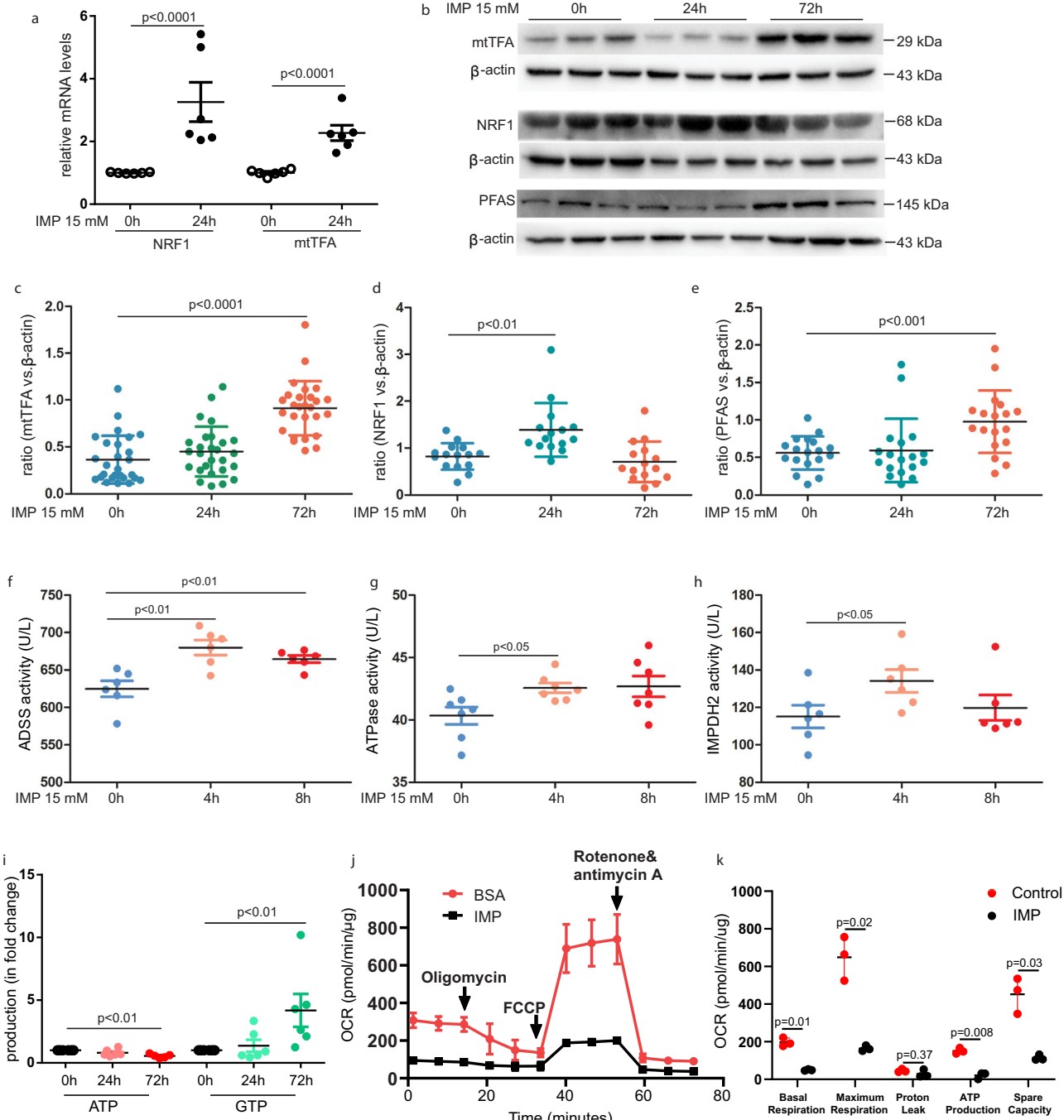

**Fig. 5 Inosine monophosphate treatment promotes mitochondrial transcription. a** RT–QPCR analysis of the mRNA expression of NRF1 and mtTFA in lineage$^{-/low}$ cells after treated with IMP for 24 h, $n = 6$. **b–e** Western blot analysis expression level of NRF1, mtTFA and PFAS in IMP-treated cells for 24 or 72 h compared with controls, $n = 15$–27. Data are presented as the means ± SEM. **f–h** ADSS, ATPase and IMPDH2 activity in lineage$^{-/low}$ cells which were stimulated with IMP for 0, 4, and 8 h, $n = 6$. **i** ATP and GTP production in lineage$^{-/low}$ cells after 24 and 72 h of IMP treatment, $n = 6$. **j, k** Oxygen consumption rate analysis of the Lineage$^{-/low}$ cells which were exposed to PBS or 15 mM IMP for 6 h, $n = 6$. One-way ANOVA with Dunnett was used to compare groups against control. IMP Inosine-5′-monophosphate, ADSS adenylosuccinate synthetase, IMPDH2 Inosine-5′-monophosphate dehydrogenase 2, ATP Adenosine triphosphate, GTP guanosine triphosphate, NRF1 nuclear respiratory factor 1, PFAS phosphoribosylformylglycinamidine synthase.

activity: $140.6 \pm 9.8$ U/L vs. $188.9 \pm 10.3$ U/L for 13 h, $57.7 \pm 5.3$ U/L vs. $90.2 \pm 8.1$ U/L for 24 h, $n = 11$–14, $p < 0.01$; plasma amylase activity: $547.7 \pm 25.2$ U/L vs. $846.3 \pm 53.4$ U/L for 13 h, $n = 11$, $p < 0.0001$; $639.0 \pm 45.6$ U/L vs. $1206.4 \pm 179.4$ U/L, $n = 11$, $p < 0.01$) (Supplementary Fig. 4b, c). When pancreatic sections were stained for H&E, we observed that administration of IMP-treated cells had accelerated the overall disease severity, as

well as the individual severity indices for each mouse (total severity score: $11.7 \pm 1.0$ vs. $20.3 \pm 0.6$, $n = 11$–14; inflammation score: $4.0 \pm 1.1$ vs. $6.2 \pm 0.8$, $p < 0.0001$; necrosis score: $2.8 \pm 0.2$ vs. $4.9 \pm 0.2$; vacuolization score: $2.1 \pm 0.3$ vs. $4.6 \pm 0.2$; edema score: $2.8 \pm 0.2$ vs. $4.7 \pm 0.1$; $n = 11$–14; $p < 0.0001$ for all) (Supplementary Fig. 4d–f). Furthermore, immunostaining of MPO within pancreatic sections confirmed the increase in inflammatory cells

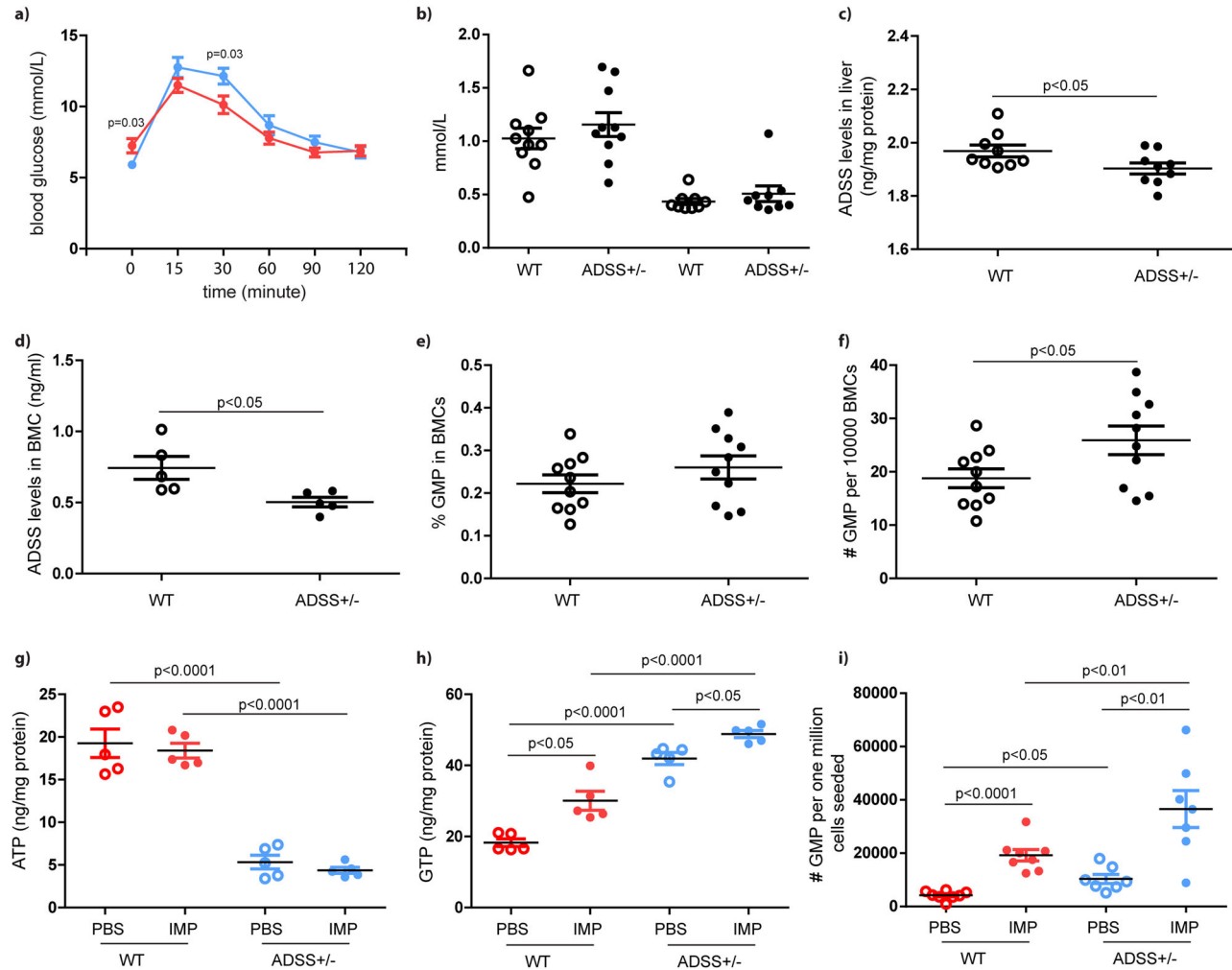

**Fig. 6 ADSS deficiency provokes more dramatic GMP cell expansion induced by IMP. a** glucose tolerance, $n = 10$. **b** Plasma levels of cholesterol and triglyceride. **c, d** ADSS expression levels in BMC expressed in liver and BMC. $n = 9$-10. **e, f** Frequency and absolute number of GMP cells in WT and ADSS$^{+/−}$ mice. $n = 10$. **g, h** ATP and GTP levels in lineage$^{−/low}$ cells after three days of IMP treatment. $n = 5$. **i** Yielded GMP cells after three days of IMP stimulation in vitro. $n = 8$. Unpaired, 2-tailed Student's $t$ test for data with normal distribution. Non-parametric Mann–Whitney analysis was used for the data did not follow normal distribution.

in mice injected with IMP-treated cells (Supplementary Fig. 4g, h). Notably, the disease severity induced by injection of IMP-treated cells was significantly attenuated in the presence of the pAkt inhibitor MK2206 (Supplementary Fig. 4b–h).

**Disturbance of purine metabolism elevates GMP cell production in ADSS$^{+/−}$ mice.** Together, these data describe a role of IMP in GMP cell expansion and myeloid cell production in vitro. We were then curious whether the disturbance of purine metabolism could modify GMP cell biology in vivo. Using spCas91.1 technology, we generated ADSS-haploinsufficient (ADSS$^{+−}$) mice and genotyped them using PCR (Supplementary Fig. 5). The fasting blood glucose levels were 5.9 mmol/L for wt mice and 7.2 mmol/L for ADSS$^{+/−}$ mice (Fig. 6a). Plasma levels of cholesterol and triglycerides did not differ between the two groups (total cholesterol: $1.03 ± 0.10$ mmol/L vs. $1.16 ± 0.11$ mmol/L, $n = 10$, $p = 0.39$; total triglyceride: $0.43 ± 0.03$ mmol/L vs. $0.51 ± 0.07$ mmol/L, $n = 9$, $p = 0.37$) (Fig. 6b). An ELISA showed that ADSS expression levels were significantly lower in the liver and BMCs of ADSS$^{+/−}$ mice compared with WT control mice (Fig. 6c, d).

The total BMC number was increased by 1.2-fold in ADSS$^{+/−}$ mice compared with wt controls ($84.6 ± 7.0 × 10^6$/mouse vs.

$99.5 ± 4.0 × 10^6$/mouse, $n = 5$, $p = 0.10$). FACS analysis showed that the frequency of lineage$^{−/low}$ cells was increased by 1.5-fold in ADSS$^{+/−}$ mice compared with a matched control ($11.2 ± 1.5\%$ vs. $7.3 ± 0.3\%$, $n = 10$, $p < 0.001$). There was no difference in GMP frequency; yet, GMP cell number was 1.4-fold higher in ADSS$^{+/−}$ mice compared with a WT control ($p = 0.04$) (Fig. 6e, f). Following 3 days of culture in vitro, IMP treatment did not appreciably alter the levels of ATP, but did result in an increase of GTP production in lineage$^{−/low}$ cells (Fig. 6g, h). FACS data showed that IMP induced a more dramatic expansion in ADSS$^{+/−}$ GMP compared with WT GMP (Fig. 6i).

**The relationship of inosine monophosphate and myeloid cell number in diabetic patients with acute pancreatitis.** Finally, we investigated whether purine metabolites affected myelopoiesis in diabetic patients with AP. Purine levels in plasma samples of patients with diabetes ($n = 50$) and/or AP ($n = 31$) were measured by metabolomics. Information on these patients is given in Table 1. Diabetic patients with AP were typically younger and presented with higher levels of total cholesterol and triglycerides, increased fasting blood glucose levels, and higher white blood cell counts in their blood compared with diabetic subjects without AP

**Table 1 General characterization of diabetic patients with or without acute pancreatitis.**

|  | DM alone | DM+AP | P value |
|---|---|---|---|
| Number in category | 50 | 31 |  |
| *All in category* |  |  |  |
| Women (0,1) | 23 (46.0%) | 9 (29.0%) | 0.13 |
| Smokers (0,1) | 18 (36.0%) | 7 (22.6%) | 0.20 |
| Hypertension (0,1) | 27 (54.0%) | 15 (48.4%) | 0.62 |
| Alcoholic intake (0,1) | 14 (28.0%) | 5 (16.1%) | 0.22 |
| Hyperlipidemia (0,1) | 20 (40.0%) | 14 (45.2%) | 0.65 |
| Cardiovascular disease (0,1) | 11 (22.0%) | 2 (6.5%) | 0.06 |
| *Mean (SD) of characteristic* |  |  |  |
| Age (years) | 58.2 ± 12.3 | 44.8 ± 17.1 | 0.0001 |
| Body mass index (kg/m$^2$) | 25.3 ± 4.0 | 27.8 ± 2.5 | 0.14 |
| Systolic blood pressure (mmHg) | 130.2 ± 22.3 | 130.9 ± 24.3 | 0.84 |
| Diastolic blood pressure (mmHg) | 79.0 ± 10.6 | 78.9 ± 14.1 | 0.95 |
| Mean arterial pressure (mmHg) | 86.5 ± 31.5 | 96.2 ± 17.1 | 0.58 |
| Heart rate (beats per minute) | 73.9 ± 18.0 | 82.8 ± 11.1 | 0.02 |
| White blood cells (x10$^9$/L) | 6.7 ± 1.7 | 12.5 ± 3.9 | <0.0001 |
| Platelets (x10$^9$/L) | 205.9 ± 71.3 | 208.6 ± 91.3 | 0.77 |
| *Biochemical data* |  |  |  |
| Serum creatinine (μmol/L) | 67.2 ± 20.8 | 66.1 ± 30.5 | 0.86 |
| Total cholesterol (mmol/L) | 4.32 ± 0.97 | 7.16 ± 3.35 | 0.0001 |
| LDL cholesterol (mmol/L) | 2.56 ± 0.82 | 1.99 ± 1.15 | 0.02 |
| HDL cholesterol (mmol/L) | 1.21 ± 0.97 | 0.95 ± 0.40 | 0.17 |
| Serum triglyceride (mmol/L) | 1.78 ± 0.90 | 13.26 ± 13.90 | 0.001 |
| Fasting plasma glucose (mmol/L) | 7.80 ± 2.62 | 12.35 ± 4.50 | 0.001 |
| HbA1c (%) | 8.73 ± 1.77 | 9.13 ± 1.72 | 0.33 |
| Uric acid (μmol/L) | 344.0 ± 74.8 | 354.7 ± 99.8 | 0.51 |
| γ-glutamyltransferase (units/L) | 42.7 ± 104.6 | 158.7 ± 194.7 | 0.0009 |

Data are expressed as mean (SD) or proportion (%).

($p \geq 0.02$ for all). Of the 31 patients, six patients were diagnosed with medium to severe AP (19.3%).

We plotted the distribution of the values measured across both groups (Fig. 7a). A volcano plot revealed the differences between each purine metabolite across the two groups (Fig. 7b). According to a Mann–Whitney U test, levels of succinyladenosine in the blood of diabetic patients with AP were decreased by 7% compared with diabetic patients without AP. Conversely, levels of 2′-deoxyguanosine 5′-monophosphate were 25.2-fold higher in diabetic patients with AP than diabetic patients without AP ($p = 0.0007$ for succinyladenosine; $p = 0.008$ for 2′-deoxyguanosine 5′-monophosphate) (Fig. 7c, d). Univariate analysis illustrates the correlations between these two purine metabolites and the proportion of granulocytes in the blood of patients in the study (Fig. 7e, f). After adjusting by age and sex, we determined that the proportion of granulocytes in the blood was increased by 2.89% (95% confidence interval (CI), −5.91, 11.69) and 2.58% (95% CI, 0.59, 4.57) for a one standard deviation increase of succinyladenosine and 2′-deoxyguanosine 5′-monophosphate, respectively ($p = 0.52$ for succinyladenosine; $p = 0.013$ for 2′-deoxyguanosine 5′-monophosphate).

## Discussion

In this study, we found that GMP cells in the BM of 24-week-old db/db mice were more proliferative and showed a greater degree of skewed myelopoiesis compared with mice at 8 weeks old, which augmented disease severity when injected into a murine model of AP. Secondly, we observed that as diabetes progressed, the microenvironment of the liver, BM, and GMP cells shifted toward enhanced purine metabolism. Thirdly, we observed that the key purine intermediate metabolite IMP was able to induce GMP cell proliferation and Gr-1$^+$ granulocyte production in vitro. Moreover, injection of IMP-treated lineage$^{-/low}$ cells potentiated disease progression in an established AP model in vivo. Fourthly, we determined that the regulation of GMP proliferation by IMP was associated with a transient increase in turnover of cKit in the plasma membrane, increased pS6 activation and effects on mitochondrial biogenesis. Lastly, we proposed a model describing the underlying mechanism behind IMP modulation of GMP cell proliferation and Gr-1$^+$ cell production. Our data suggest that Akt activation in lineage$^{-/low}$ cells is critical for the modulation of myelopoiesis by IMP, as inhibition of Akt activation diminished IMP-induced pancreatic damage following cell injection. The proposed model is shown in Fig. 7g.

Recently, an increasing number of studies have begun to characterize the roles of purine metabolites in cell biology and disease development. For example, FAMIN, an enzyme that activates adenosine deaminase, purine nucleoside phosphorylase, and methylthioadenosine phosphorylase, has been shown to suppress antigen presentation in dendritic cells and to control the antiviral T cell response. Loss of FAMIN function is the sole cause for monogenic Still's disease[14]. ATP has a short half-life and is rapidly hydrolyzed to ADP. A reduction in ATP levels is detrimental to acinar cell apoptosis. Delivery of excess ATP could therefore protect the ability of acinar cells to apoptose under conditions of AP[15]. Similarly, nicotinamide phosphoribosyltransferase, the rate-limiting enzyme in the nicotinamide adenine dinucleotide salvage pathway, strengthens macrophage polarization and inflammation in AP[16]. Here, we have uncovered a pathological role of another purine metabolite, IMP, in myelopoiesis, resulting in exaggerated inflammation and AP progression.

The progression of diabetes often induces metabolic disorders and chronic inflammation. A number of studies have shown a close link between HSPCs and GMP proliferation in BM, and skewed myelopoiesis in apoE$^{-/-}$ mice, streptozotocin-injected mice, Ins2$^{Akita}$ mice, and integrin β2$^{-/-}$ mice[9,17,18]. In agreement with these findings, our data demonstrate that skewed myelopoiesis results from enhanced GMP proliferation in diabetic db/db mice, and contributes substantially to AP progression. We further explored whether metabolites other than glucose in type 2 diabetes could regulate myelopoiesis. Using mass spectrometry, we characterized the BM microenvironment in wt and db/db mice and observed an enrichment of purine metabolites following diabetic progression. We then used a large-scale targeted metabonomic approach to uncover an acceleration of purine metabolism in FACS-sorted GMP cells and in the BM microenvironment in db/db mice at 24 weeks old compared with db/db mice at 8 weeks old. Within the purine metabolism pathways, the addition of the key nucleotide IMP initiated GMP cell expansion and mitochondrial transcription in vitro. Furthermore, injection of BMCs pre-treated with IMP reinforced disease severity in an established AP mice model in vivo. Together, these data help to understand how diabetes propagates disease progression and AP severity.

We then aimed to characterize the underlying molecular mechanism. We determined that activation of the ribosomal S6 protein was consistently identified in lineage$^{-/low}$ and GMP cells by both western blot and FACS (Fig. 4). Conversely, Akt phosphorylation was induced in lineage$^{-/low}$ but not in GMP cells, indicating that ribosomal S6 protein activation is not downstream of Akt phosphorylation in this pathway in GMP cells. Nonetheless, it is likely that IMP is able to activate Akt in other cell types and thus indirectly promote GMP cell proliferation, as inhibition of pAkt abrogated IMP-induced GMP expansion. We

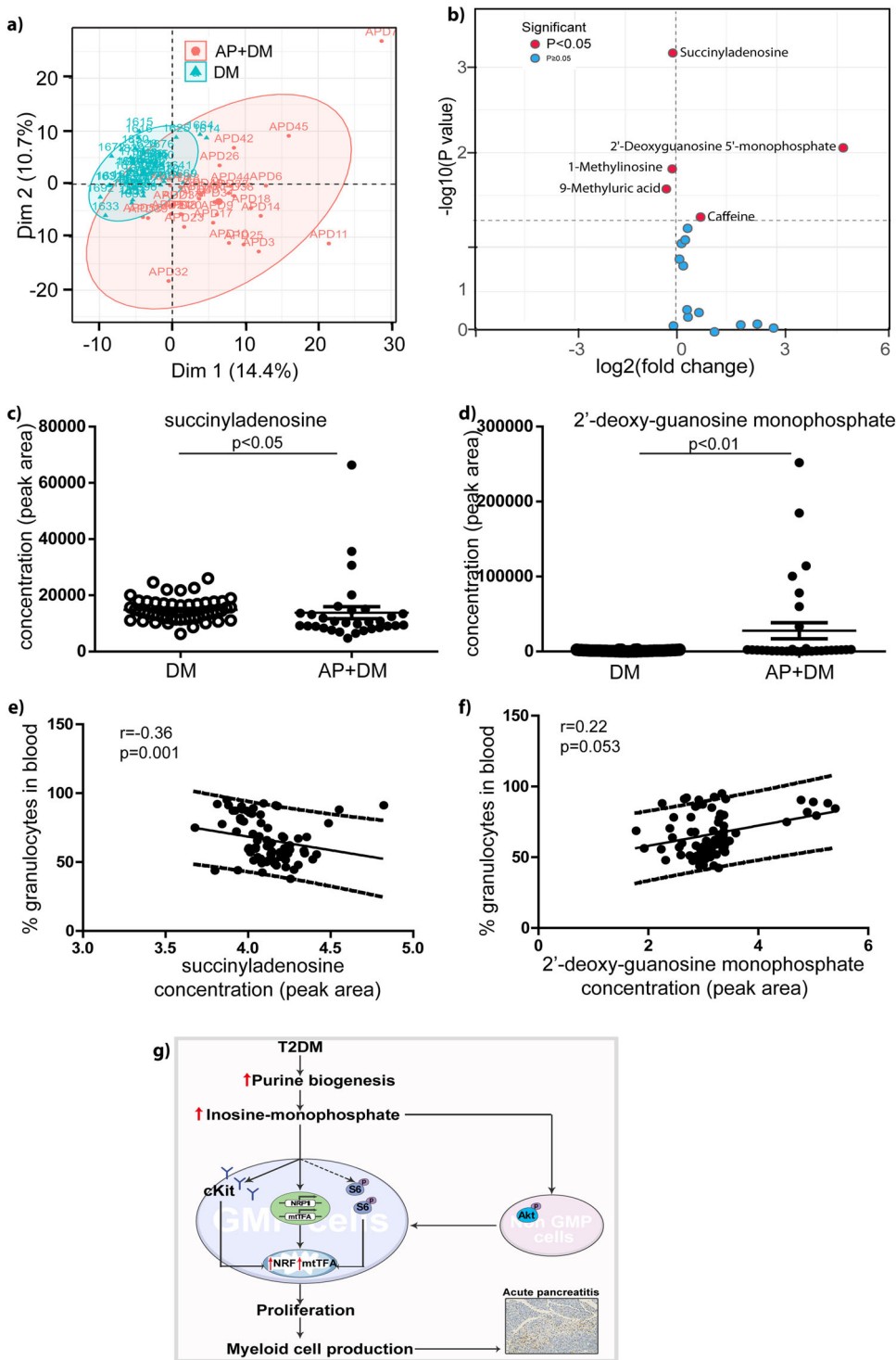

**Fig. 7 The relationship of inosine monophosphate and myeloid cell number in diabetic patient with acute pancreatitis. a** The individual plot provided the distribution of all values measured in patients with diabetes ($n = 50$) and/or AP, ($n = 31$). **b** Volcano blot presented the head-to-head comparison of each purine metabolites. **c**, **d** Succinyladenosine and 2′-Deoxyguanosine 5′-monophosphate levels in the blood of diabetic patients with AP and diabetic subjects. Unpaired, 2-tailed Student's $t$ test for data with normal distribution. Non-parametric Mann–Whitney analysis was used for the data did not follow normal distribution. $n = 31$–50. **e**, **f** Univariate analysis illustrated the opposite correlation of these two purine metabolites with granulocyte proportion in the circulation in the study populations. $n = 79$. **g** A proposed working model. Under diabetes, granulocyte-monocyte progenitor cells are proliferative, resulting in enhanced myeloid cell production. By metabolomics, purine metabolism is increased in GMP cells. The key purine metabolite, IMP, stimulates GMP cell proliferation and granulocyte production via S6 phosphorylation, cKit expression onto cell S6membrane and mitochondrial biogenesis in vitro. IMP-modulated GMP cell proliferation was through Akt activation in lineage-/low cells and inhibition of Akt activation diminished IMP-induced pancreas damage upon cell injection.

also observed that the addition of IMP to lineage$^{-/low}$ cells enhanced GTP but not ATP production. Production of AMP from IMP and AMP synthesis both require energy from guanosine triphosphate. Similarly, GMP synthesis requires energy from ATP. We then assayed enzyme activity and OCR, and determined that the activities of ADSS, ATPase and IMPDH2 were transiently increased upon treatment with IMP. This results in the consumption of ATP and the production of GTP in lineage$^{-/low}$ cells. Treatment of ADSS$^{+/-}$ BMCs with IMP resulted in an elevation of GMP cell production compared with untreated BMCs (Fig. 6). This effect was further confirmed by the relationships between succinyladenosine, 2′-deoxyguanosine 5′-monophosphate, and granulocyte proportion in diabetic patients with or without AP.

This study must be interpreted with limitations. Firstly, due to the small amount of GMP cells in mice, we could not characterize expression of the proteins mtTFA and NRF1, nor the production of ATP and GTP using current techniques. Secondly, although enriched purine metabolites were found in GMP cells by metabonomic analysis, the results of a droplet digital PCR (ddPCR) did not detect any differences in the expression of ADSS, IMPDH2, and GMP reductase between the two groups. We therefore cannot definitively conclude that a significant difference exists. It was not feasible to study protein expression in purified GMP cells. Thirdly, we do not present a mechanism for why the purine biogenesis pathway is elevated in diabetic db/db mice, as it is beyond the scope of this study. Lastly, we were unable to quantify BM-derived GMP cells, nor were we able to determine the levels of purine metabolites and nucleotides in patient GMP cells. Fourth, we could neither quantify BM-derived GMP cells nor determine purine metabolites and nucleotides in GMP cells in the patients. Lastly, we detected IMP-induced Akt activation in lineage$^{-/low}$ cells but not GMP cells. This indicates that the effect of IMP on Akt phosphorylation might be mediated via receptor expressed on lineage$^{-/low}$ but not in GMP cells. For future perspective, it would be of interest to identify the specific receptor for IMP and its expression pattern to interpret its biological function.

In summary, the key purine nucleotide IMP is not only a precursor for AMP and guanosine monophosphate synthesis, but also stimulates GMP proliferation, resulting in skewed myelopoiesis and accelerated AP progression in diabetes.

## Methods

**Human study**. The study complied with the Helsinki Declaration for the investigation of human subjects and received ethical approval from the competent Institutional Review Boards of the Beijing Youan Hospital and Capital Medical University. All the patients provided written informed consent.

**Study subjects**. Type 2 diabetic patients with or without AP were enrolled in the Department of Cardiology of Yangzhou hospital, the Department of Endocrinology of Beijing TongRen hospital, and the Department of Science and Technology of Beijing Youan hospital from March 2016 to May 2017. Diabetes was defined as either fasting plasma glucose levels ≥7.0 mmol/L or glucose levels ≥11.0 mmol/L at 2 h after an orally administered glucose load of 75 g. Hypertension was defined as blood pressure of ≥140 mmHg systolic, 90 mmHg diastolic, or the use of antihypertensive drugs. Additional characteristics including age, body weight, height, systolic blood pressure, diastolic blood pressure, heart rate, disease history and medications were recorded.

**Biochemical measurement**. After overnight fasting, venous blood samples were drawn to measure total and differential white blood cell counts, serum cholesterol and triglycerides, HDL cholesterol, HbA1c, uric acid levels, and plasma glucose levels. LDL cholesterol (LDL-c) was calculated from serum total and HDL cholesterol (HDL-c) and serum triglycerides by the Friedewald equation[19].

**Mice study**

*Mice and treatment*. The protocols for mice experiments were approved by the ethics committees of Beijing Youan Hospital. Male WT C57BL/6 J mice at 8 weeks old and obese db/db mice at 8 and 24 weeks old were used in the study. ADSS knockout is lethal; therefore, heterozygous mice were established spCas91.1 technology (ViewSolid Biotech, Beijing, China). Briefly, ADSS is predicted to

have three transcripts and exons 1–13 (AA 1-456) encode the protein. Target guides were designed for either end of region including exons 1–13 to ensure complete deletion. The resulting mRNA could not be transcribed. All the mice were maintained in the animal facility in Beijing Youan hospital, where they had free access to water and standard laboratory chow *ad libitum*. A total of 275 mice were used in the study.

To induce AP, C57BL/6 mice received ten intraperitoneal injections of 50 μg/kg caerulein at 1-hour intervals (Cae, AnaSpec, Inc. San Jose, CA, USA), as previously described[12]. Mice received the same number of PBS injection served as negative controls. To assess the effects on disease severity, BMCs of db/db mice were freshly isolated and injected intraperitoneally 1 h after the final caerulein administration. Blood samples were collected after 1- or 13-h following injection of BMCs. Mice were dissected 13 h after injection of BMCs. In the case of the pAkt inhibitor experiment, lineage$^{-/low}$ cells were isolated from WT mice and cultivated with 0- or 15- mM IMP in the presence or absence of 5 mM MK2206 for 7 days. The cells were then collected and injected into mice with caerulein-induced AP ($1 \times 10^7$ cells per recipient). The same cell injection experiments were performed using AMP-treated lineage$^{-/low}$ cells injected to AP mice.

*White blood cell counts*. Leukocytes, lymphocytes, monocytes, granulocytes, red blood cells, and blood platelets in PBS were quantified with an Auto Hematology Analyzer (Mindray, BC-2800Vet, Shanghai, China).

*Glucose tolerance test and lipid profile*. After overnight fasting, db/db mice were injected with 10% glucose (10 μL/g of body weight) intraperitoneally. Blood glucose levels were determined before injection and 15 min, 30 min, 60 min, 90 min, and 120 min after injection (Roche, USA). Blood was obtained from the tail vein after overnight fasting. Total cholesterol and triglyceride levels were measured as before using a Cholesterol kit (CHOD-PAP Method) and a Triglyceride kit (GPO-PAP Method) (BIOSINO)[6].

*Enzyme-linked immunosorbent assay (ELISA)*. The concentrations of AMP, GMP, ADSS, and IMPDH2 in BM fluid of db/db mice were measured using an ELISA. The enzymatic activity of ADSS and IMPDH2 in lineage$^{-/low}$ cells were also determined by ELISA, in accordance with the manufacturer's instructions (MLBio, Shanghai, China). Plasma lipase and amylase activities were measured using commercial kits and quantified by a Tecan Safire microplate reader as previously described[12]. Plasma levels of myeloperoxidase (MPO) were determined using a Mouse MPO ELISA Kit (Abcam, Shanghai, China).

*Quantification of ATP and GTP in liver and cultured BMCs*. BMCs and liver tissues were obtained from WT and db/db mice at 8 and 24 weeks. Equal amounts of protein were measured for GTP quantification using an enzymatic assay (ml027797, MLBio, Shanghai, China). ATP concentration was measured using a commercial Fluorometric Assay kit (213-579-1, Sigma-Aldrich, Saint Louis, US). All procedures were in accordance with the manufacturer's instructions.

**Cell signaling assay**. Equal amount of lineage$^{-/low}$ cells were treated with 15 mM IMP for 0, 15, or 30 min. Following protein extraction, the phosphorylation pathways were measure for each sample (Bio-Plex ProTM Cell Signaling Assay, Bio-Rad, USA).

*Fluorescence-activated cell sorting (FACS)*. Multicolor analysis for hematopoietic stem/progenitor cells (HSPCs), common myeloid progenitors (CMP), granulocyte-monocyte progenitors (GMP), megakaryocyte-erythroid progenitors (MEP), BMCs, and splenocytes was performed by FACS as previously described[6]. HSPCs were defined as Lin$^-$ Sca-1$^+$ cKit$^+$ cells (LSK); CMP, GMP, and MEP were defined as CD34$^+$ CD16/32$^{dim}$ cKit$^+$ Sca-1$^-$ lineage$^{-/low}$ cells, CD34$^+$ CD16/32$^+$ cKit$^+$ Sca-1$^-$ lineage$^{-/low}$ cells, and CD34$^-$ CD16/32$^{dim}$ cKit$^+$ Sca-1$^-$ lineage$^{-/low}$ cells, respectively. After red blood cell lysis, white blood cells in peripheral blood were stained with lineage allophycocyanin (APC), cKit (PE), and Sca-1 (FITC) to quantify LSK cell frequency. After treatment with inosine monophosphate, lineage$^{-/low}$ cells were harvested and stained for surface markers against GMP for FACS analysis.

To analyze the cell cycle of cultured cells, BrdU was added to a final concentration of 10 μM 18 h prior to harvesting[6]. BMCs were first stained with antibodies against GMP markers, according to the manufacturer's instructions. Cells were then permeabilized, fixed, and finally stained with FITC-conjugated BrdU prior to FACS analysis.

To study pAkt and pS6 activation, BMCs were treated with 15 mM IMP in the presence or absence of 5 mM MK2206 for 0, 15, or 30 min. After staining for GMP surface markers, cells were fixed and permeabilized by BD Cytofix/Cytoperm Fixation/Permeabilization kit following the manufacturer's instructions. The cells were then incubated with antibodies against mouse pAkt (1:50) or against mouse pS6 (1:100) for 30 min at 4 °C. and then probed with goat anti-rabbit IgG FITC (1:500) for 20 min before FACS analysis.

To quantify myeloid cell production in vitro, lineage$^{-/low}$ cells were cultured in the presence or absence of 15 mM IMP for 5 days. Cells were harvested and stained with anti-mouse Gr-1 APC for analysis by FACS.

Isotype IgG abs were used as negative controls. In total, more than 30,000 lineage$^{-/low}$ cells and 100,000 BMCs were acquired. Data were acquired on a FACSCanto II machine (BD) and analyzed using FACSDiva (BD).

FACS antibody list is shown in Supplementary Table 2.

*Cell isolation and Cell culture.* Lineage$^{-/low}$ cells were isolated from WT BMCs using the lineage cell depletion kit (Miltenyi Biotech, BergischGladbach, Germany), in accordance with the manufacturer's instructions. After isolation, lineage$^{-/low}$ cells were seeded at a density of $5 \times 10^5$ cells per well in low adsorption 24-well plates. Cells were cultured in SFEM medium (STEMCELL Technologies, Vancouver, Canada) supplemented with 10 ng/mL stem cell factor (R&D, USA), 10 ng/mL thrombopoietin (R&D, USA) and 10 ng/mL IL-3 (R&D, USA). During the study, the cells were exposed to either PBS or 15 mM IMP (MLBio, Shanghai, China) for 24 or 72 h, respectively. After harvesting, the cells were counted and stained with GMP surface markers before being subjected to analysis by FACS. For inhibitor experiments, MK2206 (MLBio, Shanghai, China) was added at a concentration of 5 mM for 1 h before the addition of IMP.

To evaluate the effects of AMP on myeloid lineage production, lineage$^{-/low}$ cells were treatment with AMP (at 0, 50 μM, or 100 μM) for 72 h, and then changes in GMP frequency were assessed by FACS. In certain experiments, cells were treated with AMP or a control for 5 days. The cells were then harvested and stained with anti-mouse CD11b, anti-mouse F4/80, and anti-mouse Gr-1 for FACS analysis.

*Proteomic sample preparation and liquid-chromatography tandem-mass spectrometry (LC-MS) analysis.* After anesthesia by intraperitoneal injection of 10% chloral hydrate, mice were perfused with saline through cardiac puncture. The BM cavity was dissected and flushed with 500 μL of sterile PBS containing 1% proteinase inhibitors. After centrifugation at 2,000 rpm for 5 min, the supernatant was collected and stored at −80 °C. Prior to mass spectrometry, the protein concentration in each sample was determined using a BCA kit (Bio-Rad). Bone marrow fluid (8 μg) was separated on a 10% SDS-PAGE gel and stained with Coomassie brilliant blue staining buffer. The lanes were then carefully cut into pieces and subjected to trypsin digestion[20].

Peptides were introduced into the mass spectrometer using Nona 415 liquid chromatography (Sciex, United States). The chromatography solvents were water/acetonitrile/formic acid (A: 98%/2%/0.1%, B: 2%/98%/0.1%). Briefly, after digestion, 2 μL of each sample was injected onto a C18 desalting column and then separated on a C18 analysis column with sequential acetonitrile gradients (from 5% to 16%, from 16% to 26%, from 26% to 40%, from 40% to 80%, and finally from 80% to 5%) at a flow rate of 0.6 μL/min. The LC-MS measurements were performed on a Triple TOF 6600 mass spectrometer (Sciex, United States) fitted with a Nanospray III source (Sciex, United States). The ion spray voltage was 2300 V, the declustering potential was 80 V, the curtain gas was at 35 psi, the nebulizer gas was at 5 psi, and the interface heater temperature was 150 °C.

**Analysis of mass spectrometry data.** Peptides present in the data were identified by matching them to the UniProt *Macaca mulatta tcheliensis* databases, producing a list of the corresponding proteins. The Paragon algorithm in Protein Pilot v 5.0 (SCIEX) was used to search the databases. Database search parameters were as follows: the trypsin used for digestion, the state of cysteine alkylation, the ID focus (typically biological modifications), and the type of search effort (typically thorough).

We first merged the two batches of mass spectrometry data to reduce the ratio of missing values. Next, the fold change (FC) of each peptide was calculated between the db24w group and the db8w group. A two-tailed Student's $t$ test was used to assess the significance of differences. We then used the criteria $FC < 0.5$ or $FC > 2.0$ and $p < 0.05$ to obtain a list of candidate peptides. For simplicity, the peptide identifiers were converted to gene names via the gene ID conversion tool on DAVID[21,22]. In addition, we performed a functional enrichment analysis of these genes using the functional annotation tool on DAVID (http://david.abcc.ncifcrf.gov/) to explore the underlying biological relevance. Gene Ontology (GO) was used to classify the differential proteins according to subcellular location and biological function. KEGG pathway analysis was performed using the KEGG database. A heat map was generated using cluster 3.0.

To gain more information, we calculated a mouse protein-protein interaction (PPI) network (BioGRID, v3.4.142)[23]. We further calculated a human PPI network, using the unweighted and weighted degrees of homology (InBioMap, v20160912)[24]. Finally, we used the Gene Importance Calculator (GIC) to gain insight into gene essentiality (bioRxiv 177923). Together, these results may facilitate further selection of genes of interest.

*Western blot analysis.* Total cellular proteins for western blot analysis were prepared and quantified by BCA Protein Assay kit (23229, Thermo). The proteins were separated by 10% sodium dodecyl sulfate-polyacrylamide gel electrophoresis (SDS-PAGE) and then transferred into a polyvinylidene difluoride (PVDF) membrane, which allowed the incubation with the appropriate antibodies.

Primary antibodies used in our experiments were as follows: Akt (cat. 9272, diluted 1:1000, CST, Danvers, MA, USA); phospho-Akt (Ser473) (D9E) (cat. 4060, diluted 1:1000, CST); S6 Ribosomal Protein (54D2) (cat. 2317, diluted 1:1000, CST); phospho-S6 Ribosomal Protein (Ser235/236) (cat. 2211, diluted 1:1000, CST); NRF1 (D9K6P) (cat. 46743, diluted 1:1000, CST); mtTFA (cat. ab272885, diluted 1:1000, Abcam, Cambridge, UK); PGC-1α (cat. 66369-1-Ig, diluted 1:1000,

Proteintech, Chicago, IL, USA); β-actin (13E5) (cat. 4970, diluted 1:2000, CST). Primary antibodies were incubated with the membrane at 4 °C overnight. The membranes were then washed and incubated with secondary antibodies (goat anti-rabbit IgG-HRP (cat. ZB-2301, diluted 1:5000, ZSGB-Bio, Beijing, China) and goat anti-mouse IgG-HRP (cat. ZB-2305, diluted 1:5000, ZSGB-Bio, Beijing, China)) conjugated to horseradish peroxidase. Immunoreactive proteins were detected by enhanced chemiluminescence (ECL). The relative density of the protein bands was quantitively determined with ImageJ software.

*RNA extraction and quantitative analysis by qPCR or ddPCR.* Lineage- Sca-1+ cKit+ (LSK) cells were prepared for quantitative PCR (qPCR). RNA was extracted from cells using TRIzol Reagent (15596018, Thermo) and total RNA was reverse transcribed into cDNA for qPCR. qPCR was conducted using AceQ Universal SYBR qPCR Master Mix (Q511-02, Vazyme, Nanjing, China). β-actin was used to normalize expression levels of genes. Primers used are shown in Supplementary Table 3).

FACS-sorted GMP cells were subjected to droplet digital PCR (ddPCR). The ddPCR was conducted on the TD-1$^{TM}$ Droplet Digital$^{TM}$ PCR System (TargetingOne, licensed in China, registration numbers: 20170025; 20190065; 20192220517) according to the manufacturer's instructions. The ddPCR reaction volume was 30 μL and contained primers, template cDNA, and Probe ddPCR SuperMix in a ddPCR platform for quantification. Droplet generation and transfer of emulsified samples to PCR plates were performed according to manufacturer's instructions (Chip Reader R1.1, TargetingOne). Primer sequences are given in Supplementary Table 4.

*Metabolomics analysis based on LC/MS.* We used targeted metabolomics to measure polar metabolites in FACS-sorted cells due to the high sensitivity of this technique. To measure polar metabolites in the plasma of human subjects, we used an untargeted metabolomics approach. For targeted metabolomics focusing on glycolytic pathways, pentose-phosphate metabolism, and tri-carboxylic acid cycle, we extracted sugars and polar metabolites using a modified version of Bligh and Dyer's protocol[25]. The upper aqueous phase was extracted, dried, and analyzed on a Thermofisher DGLC U3000 coupled with Sciex QTRAP 6500 Plus under ESI mode with the following conditions: curtain gas, 20; ion spray voltage, 5500 V; temperature, 400 °C; ion source gas 1, 35; ion source gas 2, 35[26]. Deuterated internal standards used for quantitation were d$_7$-Glucose, $^{13}C_3$-Fructose, $^{13}C_2$-Galactose, $^{13}C_{12}$-Sucose, $^{13}C_6$-Lactose, $^{13}C_{12}$-Maltose, $^{13}C_6$-UDP-Glucose, $^{13}C_6$-Glucose-6-Phosphate, $^{13}C_6$-Fructose-6-Phosphate, and $^{13}C_6$-Fructose-1,6-bisphoshate purchased from Cambridge Isotope Laboratories. For untargeted metabolomics, plasma samples were extracted with ice cold 80% methanol in water and dried in a SpeedVac. Untargeted metabolomics was conducted on an Agilent 1290 II UPLC coupled to Sciex 5600 Triple TOF Plus as previously described[27]. Metabolites were separated using an ACQUITY UPLC HSS T3 column (1.8 μm, 2.1 × 100 mm) (Waters, Dublin, Ireland) in reverse phase, and the parameters for detection were: ESI source voltage: +5.5 kV and −4.5 kV in positive and negative ion modes respectively; vaporizer temperature, 500 °C; drying gas (N2) pressure, 50 psi; nebulizer gas (N2) pressure, 50 psi; curtain gas (N2) pressure, 35 psi. The scan range was $m/z$ 60–900. Information-dependent acquisition mode was used for MS/MS analyses of the metabolites. The collision energy was set at $-35 \pm 15$ eV. Data acquisition and processing were performed using Analyst® TF 1.7.1 Software (AB Sciex, Concord, ON, Canada). All detected ions were extracted using MarkerView 1.3 (AB Sciex, Concord, ON, Canada) into Excel in the format of a two-dimensional matrix, including mass to charge ratio ($m/z$), retention time, and peak areas. Isotopic peaks were filtered. PeakView 2.2 (AB Sciex, Concord, ON, Canada) used to extract MS/MS data, and to perform comparisons with databases including the Metabolites database (AB Sciex, Concord, ON, Canada), HMDB, METLIN, and standard in-house references to annotate ion IDs.

*Determination of oxygen consumption.* Lineage$^{-/low}$ cells were treated with PBS or 15 mM IMP for 6 h. $O_2$ consumption was measured using a Seahorse Bioscience XF96 extracellular flux analyzer (XF96, Seahorse Bioscience). After measuring the basal respiration, the cells were treated with 1.5 μmol/L oligomycin for the quantification of ATP production and H$^+$ (proton) leak. The maximum respiratory capacity was then assessed by the addition of 1.5 μmol/L FCCP. Finally, mitochondrial respiration was blocked with a mixture of rotenone and antimycin A (both at 0.5 μmol/L) and the residual OCR was calculated to measure non-mitochondrial respiration.

*Histology.* Pancreatic tissue sections (4 μm thick) were obtained from a paraffin block using a rotary microtome. Hematoxylin and eosin staining was used to evaluate the severity of pancreatitis. The evaluation was performed by two investigators blind to the experiment.

*Immunohistochemistry.* To assess inflammatory status, paraffin sections were incubated with a rabbit anti-MPO polyclonal antibody (dilution: 1/1000; GB11224, Servicebio, Wuhan, China) overnight. After washing, sections were probed with horseradish peroxidase-conjugated goat anti-rabbit secondary antibody to produce a color reaction. Sections stained with a secondary antibody were used as negative controls. The results were analyzed using ImageJ software.

**Statistics and reproducibility.** Data were expressed as mean ± SEM. The SAS system (version 9.4; SAS Institute Inc., Cary, NC) was used for database

management and statistical analysis in the patient study. Myeloid cells were measured by multiplying the white blood cell count by the sum of the percentages of neutrophils and monocytes, divided by 100. Univariate analysis was performed to study the correlation of metabolite levels in blood and the number of granulocytes in the patient samples. A logarithmic transformation was performed on the metabolite values obtained by metabolomics. Confounding factors including age and sex were included in the multivariate-adjusted linear regression analysis.

In the mice study, statistical analysis was performed using GraphPad Prism (GraphPad Software Inc, La Jolla, CA, USA.). For experiments on two unpaired, normally distributed groups, we used the two-tailed Student's $t$ test. If the data from the two groups were not normally distributed, we used a non-parametric Mann–Whitney test. In experiments with three or more groups, we used a one-way ANOVA test with Dunnett when comparing treated groups against control, and ANOVA with Bonferroni was applied when comparing all groups.

A $p$ value of less than 0.05 was considered significant.

**Reporting summary**. Further information on research design is available in the Nature Research Reporting Summary linked to this article.

## Data availability

Raw data for graphs and charts are uploaded as Supplementary Data 1. Uncropped and unedited blot/ gel images are included in Supplementary Fig. 6. Further information and request for resources will be fulfilled by the Lead Contact, Dr. Ying-Mei Feng.

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

## Acknowledgements

This work was supported by the National Natural Science Foundation of China (grant numbers 81670765 and 82070841) to Y.-M.F. We thank Prof. Qinghua Cui and Dr. Pan Cui from Peking University for advice on mass spectrometry data analysis. We thank Alison Inglis, PhD, from Liwen Bianji (Edanz) (www.liwenbianji.cn) for editing the English text of a draft of this manuscript.

## Author contributions

X.-M.L., Y.D., X.-J.M., C.Y., Y.-J.Z., Y.C., and L.S. performed the molecular, cellular, and animal experiments. Y.C. carried out the human data analysis. G.L. and J.Y. collected human plasma samples and clinical data. S.M.L. performed the metabonomics experiments and analysis. G.-H.S. directed the metabonomics experiments and provided intellectual input in the study design and analysis. Y.-M.F. initiated and designed the study, directed the project, analyzed the data and prepared the manuscript.

## Competing interests

The authors declare no competing interests.
