## [Peer Review File · Communications Biology]

Reviewers' comments:

Reviewer #1 (Remarks to the Author):

Diabetes has the characteristics of hyperglycemia, metabolic disorder and chronic inflammation. In this study, Luo et al explored whether purine metabolites and how to promote skewed myelopoiesis in type-2 diabetes. Using obese db/db mice model, authors found that mice at 24 weeks old displayed increased myeloid cell count in the blood and granulocyte/monocyte progenitor (GMP) proliferation in bone marrow, compared with those aged at 8 weeks. Through targeted metabolomics, an increase in inosine monophosphate (IMP), a key purine intermediate metabolite, was found in ACS-sorted GMP of 24 week old mice. The authors further studied the regulatory mechanism of IMP on GMP proliferation and evaluated its clinical relevance in the established mouse model of acute pancreatitis. They also found a close relationship between plasma monophosphate levels and granulocyte counts in type 2 diabetes. Based on these findings, the authors concluded that purine metabolism in type 2 diabetes mellitus has changed, thereby enhancing the production of inflammatory myeloid cells. These findings extend the current knowledge of purine metabolism in the regulation of myeloid cell production, and are very interesting and highly important to the field. I think this manuscript can be taken into consideration for acceptance in the journal of Communication Biology after addressing the following issues:

1. AMP an important regulator of AMPK pathway, which has been involved in the pathogenesis of diabetes and diabetic complications. Does AMP also play a role in AP? How to distinguish the contribution of the AMP and IMP to the pathogenesis of the AP since IMP is a precursor for AMP.
2. Please check language carefully. For instance, Page 41, Line 884 "inosine phosphate" should be "inosine monophosphate", 2. Page 17 Line 395, Page 45, Line 928, "2'-Deoxyguanosine 5'-monophosphate" should be "2'-deoxyguanosine 5'-monophosphate".

Reviewer #3 (Remarks to the Author):

This manuscript investigates the link between metabolic changes and the pro-inflammatory myelopoiesis that occurs during the progression of diabetes and how this contributes to severity of acute pancreatitis. Specifically, this study demonstrates that bone marrow-derived granulocyte/macrophage progenitor (GMP) cells exhibited a proliferative phenotype characteristic of myelopoiesis during the course of type 2 diabetes (i.e. at 24 weeks vs 8 weeks in db/db mice). This heightened pro-inflammatory landscape also contributed to the severity of experimentally-induced acute pancreatitis (AP). The authors then went on to investigate the molecular mechanism for these changes using a variety of powerful techniques, including cellular/biochemical assays, in vivo models and human clinical studies, which focussed on dissecting the full metabolic profile of these bone-marrow-derived GMP cells. They discovered that inosine monophosphate (IMP), a product of purine metabolism, was responsible for the pro-inflammatory myelopoiesis that involved the phosphorylation and thus activation of Akt. Consequently, injection of control mice with IMP-treated GMP cells enhanced the severity of AP, that was attenuated by an Akt inhibitor.

This study provides a comprehensive understanding of the metabolic changes during the course of diabetes that heightens the pro-inflammatory landscape and thus contributes to severe acute pancreatitis. This offers a variety of novel potential therapeutic interventions for the treatment of diabetes and/or acute pancreatitis. However, there are a few important issues that should be dealt with before this is suitable for publication.

- 1-There are some important controls missing from the caerulein-induced acute pancreatitis experiments (Fig 1). It is the convention when performing these in vivo pancreatitis experiments to compare the various readouts of pancreatitis between mice receiving caerulein (10 hourly injections) vs those receiving the same number of saline injections (PBS). This provides the baseline or "normal" functional readout for all established readouts of pancreatitis (plasma amylase/lipase, H&E severity scores, MPO immunohistochemistry) and controls for multiple injections (independent of caerulein). Instead, the authors compare two groups of mice, both receiving caerulein to induce pancreatitis, and both injected with bone marrow-derived cells (BMCs) from diabetic db/db mice that were either 8 weeks or 24 weeks old. No groups of mice

received PBS. The injection of BMCs, the age and diabetic status of the mice from which the injected BMCs were derived from are independent variables that need to be controlled. Therefore, additional control groups include mice injected with BMCs from normal WT mice that are 8 weeks vs 24 weeks old.

2-Similar PBS injection controls are missing from Supplementary figure 3, in which mice were injected with naïve GMP cells that have been treated with either PBS, IMP or IMP plus Akt inhibitor. At least with these experiments there was an appropriate control, as mice were injected with PBS-treated GMP cells, which is much better than Fig 1 (main manuscript).

I understand that there is increasing pressure to reduce the number of animals and unnecessary animal suffering, but the omission of important control groups like these means that the reader is being asked to interpret differences in pancreatitis severity in various caerulein-treatment groups without knowing what "normal" looks like. The authors should either include additional and appropriate control groups or provide a thorough justification as to why these control groups were not included otherwise this casts some doubt over the overall interpretation.

Minor changes

-A similar time course explaining the experimental design that is used in Supplementary Fig 3 should be used in Fig 3.

-Could the authors provide the statistical test used for each parameter measured (where p values are provided) in the legend text of each figure? There are numerous p values quoted throughout the results description but not always with a description of the statistical test. In fact it might help with the flow of the narrative if the p values were provided in the legend text rather than the Results description.

-There are numerous acronyms that are not defined and terminology not explained throughout the manuscript. Although most acronyms are defined in the Abbreviations list, it would be very helpful for the flow of narrative if these were defined or explained the first time they are used.

Examples include:

SR-BI^{-/-} and apoE^{-/-}

Lineage Sca-1⁺cKit⁺

Lineage⁻/low cells

-Line 106-108 defines a hypothesis that does not relate to acute pancreatitis.

-For the Seahorse experiments, why was 6 hours incubation chosen rather than shorter periods, given that many other assays used an incubation period of 15-30 minutes (e.g. pAkt western blot)?

-Line 265-268 frequently uses the phrase pAkt activation. Phosphorylation of Akt leads to Akt activation and is thus a readout of Akt activation.

-Line 276 and 287, replace "accelerate" with "enhance" or "potentiate". Accelerate has a time component.

-Could the authors speculate as to the mechanism for IMP-induced Akt phosphorylation?

Reviewers' comments:

Reviewer #1 (Remarks to the Author):

Diabetes has the characteristics of hyperglycemia, metabolic disorder and chronic inflammation. In this study, Luo et al explored whether purine metabolites and how to promote skewed myelopoiesis in type-2 diabetes. Using obese db/db mice model, authors found that mice at 24 weeks old displayed increased myeloid cell count in the blood and granulocyte/monocyte progenitor (GMP) proliferation in bone marrow, compared with those aged at 8 weeks. Through targeted metabolomics, an increase in inosine monophosphate (IMP), a key purine intermediate metabolite, was found in ACS-sorted GMP of 24 week old mice. The authors further studied the regulatory mechanism of IMP on GMP proliferation and evaluated its clinical relevance in the established mouse model of acute pancreatitis. They also found a close relationship between plasma monophosphate levels and granulocyte counts in type 2 diabetes. Based on these findings, the authors concluded that purine metabolism in type 2 diabetes mellitus has changed, thereby enhancing the production of inflammatory myeloid cells. These findings extend the current knowledge of purine metabolism in the regulation of myeloid cell production, and are very interesting and highly important to the field. I think this manuscript can be taken into consideration for acceptance in the journal of Communication Biology after addressing the following issues:

1. AMP an important regulator of AMPK pathway, which has been involved in the pathogenesis of diabetes and diabetic complications. Does AMP also play a role in AP? How to distinguish the contribution of the AMP and IMP to the pathogenesis of the AP since IMP is a precursor for AMP.

Response: We appreciate the reviewer's comment. In the rebuttal, we performed experiments to assess the effect of AMP on granulocyte/monocyte progenitor (GMP) proliferation and myeloid cell production. Thereafter, Lineage^{-low} cells treated with or without AMP were injected to wild-type mice with established acute pancreatitis to

assess disease severity by plasma lipase and amylase activity and H&E staining.

In the Method, on page 24, we added “To evaluate the effects of AMP on myeloid lineage production, lineage^{-low} cells were treatment with AMP (at 0, 50 μ M, or 100 μ M) for 72 hours, and then changes in GMP frequency was assessed by FACS. In certain experiments, cells were treated with AMP or a control for 5 days. The cells were then harvested and stained with anti-mouse CD11b, anti-mouse F4/80, and anti-mouse Gr-1 for FACS analysis” On page 21, we added “The same cell injection experiments were performed using AMP-treated lineage^{-low} cells injected to AP mice.”

In the Result, on page 11, we added “Adenosine monophosphate (AMP) is an important regulator of the 5' AMP-activated protein kinase pathway, which has been implicated in the pathogenesis of diabetes and diabetic complications. To determine whether AMP has any effect on GMP proliferation, lineage^{-low} cells were treated with different amounts of AMP for 3 days. Using FACS, we determined that neither GMP frequency nor myeloid cell production was altered by AMP treatment ($p \geq 0.47$) (Supplementary Fig. 2a, b). Next, we treated lineage^{-low} cells with either PBS or 100 μ M AMP for 7 days and then injected the cells intraperitoneally into wt recipients with established AP. AP mice that did not receive a cell injection were used as a control. Following cell injection, we observed that the activity levels of plasma amylase and lipase were similar between all groups ($p \geq 0.10$ for all) (Supplementary Fig. 2c, d). Consistently, both the overall severity and each individual severity score were similar in AP mice that had or had not received a cell injection ($p = 0.42$ for total severity score; $p \geq 0.25$ for individual severity scores) (Supplementary Fig. 2e–g).”

2. Please check language carefully. For instance, Page 41, Line 884 “inosine phosphate” should be “inosine monophosphate”, 2. Page 17 Line 395, Page 45, Line 928, “2'-Deoxyguanosine 5'-monophosphate” should be “2'-deoxyguanosine 5'-monophosphate”.

Response: We checked the text and corrected the errors thoroughly. The revised manuscript was edited by professional to improve the language quality as well.

Reviewer #3 (Remarks to the Author):

This manuscript investigates the link between metabolic changes and the pro-inflammatory myelopoiesis that occurs during the progression of diabetes and how this contributes to severity of acute pancreatitis. Specifically, this study demonstrates that bone marrow-derived granulocyte/macrophage progenitor (GMP) cells exhibited a proliferative phenotype characteristic of myelopoiesis during the course of type 2 diabetes (i.e. at 24 weeks vs 8 weeks in db/db mice). This heightened pro-inflammatory landscape also contributed to the severity of experimentally-induced acute pancreatitis (AP). The authors then went on to investigate the molecular mechanism for these changes using a variety of powerful techniques, including cellular/biochemical assays, in vivo models and human clinical studies, which focussed on dissecting the full metabolic profile of these bone-marrow-derived GMP cells. They discovered that inosine monophosphate (IMP), a product of purine metabolism, was responsible for the pro-inflammatory myelopoiesis that involved the phosphorylation and thus activation of Akt. Consequently, injection of control mice with IMP-treated GMP cells enhanced the severity of AP, that was attenuated by an Akt inhibitor.

This study provides a comprehensive understanding of the metabolic changes during the course of diabetes that heightens the pro-inflammatory landscape and thus contributes to severe acute pancreatitis. This offers a variety of novel potential therapeutic interventions for the treatment of diabetes and/or acute pancreatitis. However, there are a few important issues that should be dealt with before this is suitable for publication.

1-There are some important controls missing from the caerulein-induced acute pancreatitis experiments (Fig 1). It is the convention when performing these in vivo pancreatitis experiments to compare the various readouts of pancreatitis between mice receiving caerulein (10 hourly injections) vs those receiving the same number of saline

injections (PBS). This provides the baseline or “normal” functional readout for all established readouts of pancreatitis (plasma amylase/lipase, H&E severity scores, MPO immunohistochemistry) and controls for multiple injections (independent of caerulein). Instead, the authors compare two groups of mice, both receiving caerulein to induce pancreatitis, and both injected with bone marrow-derived cells (BMCs) from diabetic db/db mice that were either 8 weeks or 24 weeks old. No groups of mice received PBS. The injection of BMCs, the age and diabetic status of the mice from which the injected BMCs were derived from are independent variables that need to be controlled. Therefore, additional control groups include mice injected with BMCs from normal WT mice that are 8 weeks vs 24 weeks old.

Response: We appreciate the reviewer’s comments. Caerulein injections are frequently used to induce acute pancreatitis (AP) in mice. Previously, Gao *et al* reported that db/db mice developed severe acinar damage than wild-type controls after caerulein injection (Gao et al, Am J Transl Res 2018). In line with that, in our previous study, we showed that 10 hourly caerulein injection induced severe and reproducible AP injury, when taking non-injected mice as controls. We further showed that injection of diabetic bone marrow cells promoted more severe AP injury compared with mice received caerulein injection alone or followed by bone marrow cell administration (Luo et al, J Immunol Res 2021).

In this rebuttal, we repeated AP experiments and included mice receiving 10 hourly injections of PBS as controls. The representative H&E images were shown in Supplementary Fig2g. In the revised manuscript, on page 18, we added “Mice received the same number of PBS injection served as negative controls” In the Method. The representative H&E images were shown in Supplementary Fig.2g.

2-Similar PBS injection controls are missing from Supplementary figure 3, in which mice were injected with naïve GMP cells that have been treated with either PBS, IMP or IMP plus Akt inhibitor. At least with these experiments there was an appropriate control, as mice were injected with PBS-treated GMP cells, which is much better that

Fig 1 (main manuscript).

Response: In this rebuttal, we repeated AP experiments and included mice receiving 10 hourly injections of PBS as controls. The representative H&E images were shown in Supplementary Fig2g. In the revised manuscript, on page 18, we added “Mice received the same number of PBS injection served as negative controls” In the Method. As shown in Supplementary Fig.2g, no injury sign was detected in the mice receiving PBS injections and therefore, the total and individual severity scores were zero. Based on the findings above, in Supplementary Figure 3, wild-type mice with induced AP were received lineage^{-low} cells treated with PBS, MK2206, IMP, or IMP plus MK2206.

I understand that there is increasing pressure to reduce the number of animals and unnecessary animal suffering, but the omission of important control groups like these means that the reader is being asked to interpret differences in pancreatitis severity in various caerulein-treatment groups without knowing what “normal” looks like. The authors should either include additional and appropriate control groups or provide a thorough justification as to why these control groups were not included otherwise this casts some doubt over the overall interpretation.

Response: We appreciate the reviewer’s comments and advice. Indeed, caerulein injections are frequently used to induce acute pancreatitis (AP). Previously, Gao *et al* reported that db/db mice developed severe acinar damage than wild-type controls after caerulein injection (Gao et al, Am J Transl Res 2018). In line with that, we showed that 10 hourly caerulein injection induced severe and reproducible AP injury, taking non-injected mice as controls. We further showed that injection of diabetic BMC promoted more severe AP injury compared with mice received caerulein injection alone or followed by BMC administration. In the experiments, non-injected wild-type mice served as controls (Luo et al, J Immunol Res 2021).

In this rebuttal, we repeated AP experiments and included mice receiving 10 hourly injections of PBS as controls. The representative H&E images were shown in Supplementary Fig2g. In the revised manuscript, on page 18, we added “Mice received

the same number of PBS injection served as negative controls” In the Method. The representative H&E images were shown in Supplementary Fig.2g.

Minor changes

-A similar time course explaining the experimental design that is used in Supplementary Fig 3 should be used in Fig 3.

Response: A scheme of experimental design was added in Fig 3.

-Could the authors provide the statistical test used for each parameter measured (where p values are provided) in the legend text of each figure? There are numerous p values quoted throughout the results description but not always with a description of the statistical test. In fact it might help with the flow of the narrative if the p values were provided in the legend text rather than the Results description.

Response: Statistical method were added to each figure legend. Precise P value was added to each figure for better reading.

-There are numerous acronyms that are not defined and terminology not explained throughout the manuscript. Although most acronyms are defined in the Abbreviations list, it would be very helpful for the flow of narrative if these were defined of explained the first time they are used.

Examples include:

SR-BI-/- and apoE-/-

Lineage Sca-1+cKit+

Lineage-/low cells

Response: We provided the definition in the rebuttal.

-Line 106-108 defines a hypothesis that does not relate to acute pancreatitis.

Response: There were two aims in the study. First, we aimed to identify the metabolites rather than glucose that were altered in T2DM diabetes and modulated myeloid progenitor cell proliferation and skewed myelopoiesis. And second, we aimed to explore the pathological meaning behind the skewed myelopoiesis induced by the candidate metabolite in T2DM. For a better understanding, we rewrote this sentence as following: “In this study, we interrogated the role of other metabolites that result from the altered metabolic state in T2DM diabetes, and investigated whether other metabolites could also modulate the proliferation of myeloid progenitor cells, resulting in myelopoiesis and strengthened inflammation in diabetic vascular complication.”

-For the Seahorse experiments, why was 6 hours incubation chosen rather than shorter periods, given that many other assays used an incubation periods of 15-30 minutes (e.g. pAkt western blot)?

Response: We appreciate the reviewer’s comment on that. It is one of the key questions that we would like to investigate. In the current study, cells were incubated 6 hours before oxygen consumption measurement, which was decided based on the results of ADSS activity and ATPase activity assays. For future perspective, it would be great to determine ATP production kinetics in granulocyte/monocyte progenitors.

-Line 265-268 frequently uses the phrase pAkt activation. Phosphorylation of Akt leads to Akt activation and is thus a readout of Akt activation.

Response: We fully agree with the reviewer. In the rebuttal, the phrase “pAkt activation” was replaced either by “phosphorylation of Akt” or by “Akt activation”.

-Line 276 and 287, replace “accelerate” with “enhance” or “potentiate”. Accelerate has a time component.

Response: The text was modified as suggested.

-Could the authors speculate as to the mechanism for IMP-induced Akt phosphorylation?

Response: Yes, this is another question that we would pursue in the future. In the discussion, we added “Lastly, we detected IMP-induced Akt activation in lineage^{-low} cells but not GMP cells. This indicates that the effect of IMP on Akt phosphorylation might be mediated via receptor expressed on lineage^{-low} but not in GMP cells. For future perspective, it would be of interest to identify the specific receptor for IMP and its expression pattern to interpret its biological function.”

REVIEWERS' COMMENTS:

Reviewer #1 (Remarks to the Author):

The revised manuscript is well corrected, detailed analyses and more supporting information have been added in the context, which make the whole manuscript expressing more clearly. The language has been proofread by professional language editing company. Thus, I suggest the manuscript can be accepted by the Communications Biology journal.

Reviewer #3 (Remarks to the Author):

All issues from the originally submitted manuscript have been resolved to my satisfaction in the revised version. I would like to congratulate the authors on a fine paper.